# Real-time detection of highly oxidized organosulfates and BSOA marker compounds during the F–BEACh 2014 field study

Martin Brüggemann[1,2,‡], Laurent Poulain[3], Andreas Held[4], Torsten Stelzer[1], Christoph Zuth[1], Stefanie Richters[3], Anke Mutzel[3], Dominik van Pinxteren[3], Yoshiteru Iinuma[3], Sarmite Katkevica[4], René Rabe[3], Hartmut Herrmann[3], and Thorsten Hoffmann[1]

[1]Institute of Inorganic and Analytical Chemistry, University of Mainz, Duesbergweg 10–14, 55128 Mainz, Germany
[2]Max Planck Graduate Center, Staudinger Weg 9, 55128 Mainz, Germany
[3]Leibniz-Institut für Troposphärenforschung (TROPOS), Permoserstr. 15, 04318 Leipzig, Germany
[4]University of Bayreuth, Atmospheric Chemistry, Dr.-Hans-Frisch-Straße 1–3, 95448 Bayreuth, Germany
[‡]now at: CNRS, UMR5256, IRCELYON, Institut de Recherches sur la Catalyse et l'Environnement de Lyon, Villeurbanne F-69626, France

Correspondence to: T. Hoffmann (t.hoffmann@uni-mainz.de)

**Abstract.** The chemical composition of ambient organic aerosols was analyzed using complementary mass spectrometric techniques during a field study in Central Europe in July 2014 (Fichtelgebirge – Biogenic Emission and Aerosol Chemistry, F–BEACh 2014). Among several common biogenic secondary organic aerosol (BSOA) marker compounds, 93 acidic oxygenated hydrocarbons were detected with elevated abundances and were, thus, attributed to be characteristic for the organic aerosol mass at the site. Monoterpene measurements exhibited median mixing ratios of 1.6 ppb$_\text{V}$ and 0.8 ppb$_\text{V}$ for in and above canopy levels, respectively. Nonetheless, concentrations for early-generation oxidation products were rather low, e.g. pinic acid ($c = 4.7$ ($\pm 2.5$) ng·m$^{-3}$). In contrast, high concentrations were found for later-generation photooxidation products such as 3-methyl-1,2,3-butanetricarboxylic acid (MBTCA, $c = 13.8$ ($\pm 9.0$) ng·m$^{-3}$) and 3-carboxyheptanedioic acid ($c = 10.2$ ($\pm 6.6$) ng·m$^{-3}$), suggesting that aged aerosol masses were present during the campaign period. In agreement, HYSPLIT trajectory calculations indicate that most of the arriving air masses traveled long distances (>1,500 km) over land with high solar radiation

In addition, around 47% of the detected compounds from filter sample analysis were sulfur-containing, confirming a rather high anthropogenic impact on biogenic emissions and their oxidation processes. Among the sulfur-containing compounds, several organosulfates, nitrooxy organosulfates, and highly oxidized organosulfates (HOOS) were tentatively identified by high resolution mass spectrometry. Correlations among HOOS, sulfate and highly oxidized multifunctional organic compounds (HOMs) support the hypothesis of previous studies that HOOS are formed by reactions of gas-phase HOMs with particulate sulfate. Moreover, periods with high relative humidity indicate that aqueous-phase chemistry might play a major role in HOOS production. However, for dryer periods, coinciding signals for HOOS and gas-phase peroxyradicals (RO$_2$$^\bullet$) were observed, suggesting RO$_2$$^\bullet$ to be involved in HOOS formation.

# 1 Introduction

Secondary organic aerosols (SOAs) are a major component of tropospheric particulate matter and known to affect the Earth's climate as well as human health (Pöschl, 2005; Baltensperger et al., 2008; Hallquist et al., 2009; Intergovernmental Panel on Climate Change, 2014; Nozière et al., 2015). In general, SOA is formed by phase transition of oxidation products of volatile
organic compounds (VOCs). Depending on the source of these VOCs the resulting SOA can be classified as anthropogenic SOA (ASOA), e.g. from fossil fuel combustion, or biogenic SOA (BSOA), e.g. from terrestrial or marine ecosystems (Hallquist et al., 2009; Nozière et al., 2015). Globally, BSOA is expected to dominate the annual mass budget of SOA to a large extent (Henze et al., 2008; Hallquist et al., 2009), although it was shown that regionally ASOA can represent the main fraction of aerosol mass (Aiken et al., 2009; Fushimi et al., 2011).

In the past, several marker compounds were discovered which often allow a source apportionment and, hence, a differentiation between ASOA and BSOA. As recently reported, organic acids can account for up to 51% of the OA mass in coniferous forest regions (Yatavelli et al., 2015). In agreement, common BSOA marker compounds for monoterpenes mostly comprise carboxylic acids and corresponding derivates, such as pinic acid (Yu et al., 1998; Hoffmann et al., 1998), 2-hydroxyterpenylic acid (Claeys et al., 2009) or diaterpenylic acid acetate (Iinuma et al., 2009; Yasmeen et al., 2011). These oxidation products
are formed by reactions of VOCs with atmospheric oxidants such as ozone, OH radicals or $NO_3$ radicals and are ideally characteristic for their precursor VOC. Moreover, oxidation products such as 3-methyl-1,2,3-butanetricarboxylic acid (MBTCA) are formed by photochemical oxidation of earlier-generation marker compounds, thus, allowing to trace chemical ageing of SOA in the atmosphere (Szmigielski et al., 2007; Müller et al., 2012). In addition to these solely carbon-, hydrogen-, and oxygen-containing compounds (CHO), the class of organosulfates (OS) and nitrooxy organosulfates (NOS) is almost
ubiquitously found in SOA particles, exhibiting supplementary marker compounds for VOC precursors. However, OS and NOS compounds might represent oxidation products of biogenic VOCs in anthropogenically influenced air masses (Zhang et al., 2009; Goldstein et al., 2009; Kristensen and Glasius, 2011). Studies have shown that OS and NOS are formed in the condensed phase, either from VOC gas-phase oxidation products with sulfuric acid in acidic sulfate aerosols (Iinuma et al., 2005; Liggio and Li, 2006; Iinuma et al., 2007; Surratt et al., 2007; Surratt et al., 2008; Shalamazari et al., 2014; Shalamazari
et al., 2016), or also directly by the reaction of gaseous $SO_2$ with unsaturated carboxylic acids (Shang et al., 2016, Passananti et al., 2016). Furthermore, it was shown that such compounds are formed by nucleophilic substitution of nitrate groups by sulfate (Darer et al., 2011; Hu et al., 2011), or by heterogeneous chemistry of gas-phase organic hydroperoxides, which might undergo acid-catalyzed perhydrolysis followed by reaction with sulfate ions (Riva et al., 2016a, 2016b). Moreover, radical mechanisms involving photochemically generated sulfate radicals might represent an additional formation pathway in aerosol
particles at neutral pH (Nozière et al., 2010; Schindelka et al., 2013).

Lately a new class of monoterpene oxidation products in the gas-phase was described, named highly oxidized multifunctional organic compounds (HOMs) (sometimes also referred to as extremely low volatile organic compounds, ELVOCs) (Ehn et al., 2012; Ehn et al., 2014). These compounds exhibit O/C ratios of 0.5–1.1 and, thus, should contain several functional groups,

possibly decreasing their vapor pressures to ranges which are significantly lower than for typical BSOA marker compounds (Ehn et al., 2014). Nonetheless, recently it was shown that HOM monomers formed from the oxidation of α-pinene are unlikely to exhibit saturation vapor pressures in the range of ELVOCs – even when their O:C ratios are close to 1 (Kurten et al., 2016). Since HOM formation is explained by auto-oxidation processes, it is expected that multiple hydroperoxide groups are typically present per molecule (Crounse et al., 2013; Ehn et al., 2014). In agreement with this auto-oxidation hypothesis, Mutzel et al. (2015) recently showed that several HOMs contain at least one carbonyl group within their structure. Although a comprehensive structural elucidation of HOMs was not possible until now, it is assumed that these compounds largely contribute to both particle formation and growth (Riipinen et al., 2011; Donahue et al., 2012; Zhao et al., 2013, Tröstl et al., 2016).

Although the existence of HOMs was clearly demonstrated several times from gas-phase measurements (Ehn et al., 2012; Ehn et al., 2014; Rissanen et al., 2014; Jokinen et al., 2015; Mentel et al., 2015; Mutzel et al., 2015), their fate after phase transition still remains quite unclear. It has been hypothesized that due to the presence of hydroperoxide groups HOMs might participate in accretion reactions (Hallquist et al., 2009; Shiraiwa et al., 2013) or decompose via the Korcek mechanism (Mutzel et al., 2015), resulting in the formation of carboxylic acids, including common BSOA marker compounds. Furthermore, from recent measurements it was speculated that the simultaneous presence of gas-phase HOMs and particulate sulfate might lead to the formation of highly oxidized organosulfates (HOOS), i.e. organosulfates with O/C ratios >1.0 (Mutzel et al., 2015), although evidence for this hypothesis is rather unsatisfactory since it is mainly based on model calculations and offline measurements. Nonetheless, recently Riva et al. (2016a) reported on the formation of organosulfates from the acid catalyzed hydrolysis of isoprene-derived organic hydroperoxides.

In this study, several state-of-the-art mass spectrometric techniques were used in a complementary approach to characterize the organic aerosol fraction at a rural field site in Central Europe during summer 2014. The applied techniques comprise the recently described Aerosol Flowing Atmospheric-Pressure Afterglow Mass Spectrometry (AeroFAPA–MS) (Brüggemann et al., 2015), high resolution time-of-flight Aerosol Mass Spectrometry (AMS) (Canagaratna et al., 2007), and Chemical Ionization Atmospheric-Pressure interface Time-of-Flight Mass Spectrometry (CI-APi-TOFMS) using nitrate ($NO_3^-$) as ionization reagent (Jokinen et al., 2012). Furthermore, the detection of acidic organic compounds, such as carboxylic acids and OS, was extended by non-target analysis of filter samples using High Resolution Mass Spectrometry (HRMS) in combination with Ultra-High Pressure Liquid Chromatography (UHPLC). Besides the detection of common BSOA marker compounds, the formation of HOOS and their correlation to HOMs was investigated using online and offline instrumentation.

## 2 Experimental

### 2.1 Field Site Description

All measurements were conducted in July 2014 (15[th]–27[th]) during the F–BEACh 2014 (Fichtelgebirge - Biogenic Emissions and Aerosol Chemistry) field campaign. The measurement site was located in a rural area at an altitude of 766 m a.s.l. in the

Fichtelgebirge mountain range in Southeast Germany (BayCEER Waldstein-Pflanzgarten, 50°08'35" N, 11°51'49" E, operated by the University of Bayreuth). The site is surrounded by a mostly coniferous forest which is dominated by Norway spruce (~90%). The canopy height and displacement height are ~23 m and ~15 m, respectively. A mixture of larch, beech, maple, and pine accounts for the rest of the tree population (Staudt and Foken, 2007). All instruments were arranged closely with inlet heights of 4–6 m above ground and at a distance of less than 10 m. Solely, VOC cartridges were sampled in and above canopy level at a distance of ~200 m from the other instruments.

## 2.2 AeroFAPA–MS Measurements

The AeroFAPA ion source was used in combination with an ion trap mass spectrometer (LCQ Deca XP Plus, Thermo, San José, CA, USA) for real-time analysis of ambient organic aerosol particles. Since a detailed description of the technique can be found elsewhere (Brüggemann et al., 2015), only a brief description will be given here. In general, AeroFAPA–MS is a soft-ionization technique which allows the online detection of organic compounds in aerosol particles. The negative mode, which was applied throughout the field study, is selective towards acidic compounds, such as carboxylic acids and (nitrooxy) organosulfates. For the analysis, aerosol particles were drawn from a height of ~4 m above ground into the manifold of the AeroFAPA at a flow rate of 0.9 L min$^{-1}$. Before reaching the AeroFAPA–MS, the aerosol stream passed an activated charcoal denuder in order to remove gaseous species from the aerosol sample. Evaporation of organic aerosol components prior to ionization was supported by heating the inlet to 200 °C. Although heating is a common approach for aerosol evaporation and analysis, it should be noted here that recently Lopez-Hilfiker et al. (2016) reported on the decomposition of accretion products upon heating. Therefore, contributions of such decomposition products cannot be completely ruled out here. A helium glow discharge plasma was used to generate excited helium atoms and primary reagent ions which ionized the compounds of interest in the so-called afterglow region. During the campaign, a current of 55 mA was used, resulting in a discharge voltage of ~400 V. In addition, a potential of –15 V was applied to the exit capillary of the AeroFAPA to enhance ion transmission. The resulting analyte ions, typically [M–H]$^-$, were then sampled and detected by the mass spectrometer. Fragmentation and adduct formation was only observed to a minor extent under laboratory conditions (<5% of signal for [M–H]$^-$), however, such processes might have influenced the observed signals to a greater extent during the field campaign due to ambient conditions. A voltage of –15 V was applied to the mass spectrometer inlet capillary, equaling the potential of the AeroFAPA's exit capillary. The tube lens was held at 0 V. All mass spectra were recorded in automatic gain-control mode with 300 microscans spectrum$^{-1}$, giving roughly one full scan mass spectrum (*m/z* 130–500) per minute. The maximum ion trap injection time was set to 200 ms. MS$^n$ experiments were performed to elucidate the structure of the detected compounds. Data were recorded using XCalibur 2.0.7. Background subtraction of the acquired mass spectra was conducted by measuring a blank sample for half an hour every day. For the subsequent data analysis all files were converted to text files and analyzed using Matlab (R2014b, Mathworks Inc., USA). In order to compare and correlate data from different instruments a unified time vector was created with time intervals of 10 minutes. Thus, all signals, except the filter sample data, are average values for 10 minutes.

**2.3 AMS Measurements**

A High Resolution Time-of-Flight Aerosol Mass Spectrometer (HR–ToF–AMS, Aerodyne, USA, Canagaratna et al., 2007), was used to measure the submicron mass concentrations and size distributions of non-refractory particulate organic matter, sulfate, nitrate, ammonium and chloride. The AMS was located in an adjacent laboratory container and connected to a sampling line with a $PM_{10}$ inlet located at ~6 m above ground level. Relative humidity on the sampling line was maintained below 35% using a Nafion® dryer. A chemical dependent collection efficiency (CDCE) was applied on the AMS data according to Middlebrook et al. (2012). The quality control of the data acquired by the AMS was made according to Poulain et al. (2014).

**2.4 CI-APi-TOFMS Measurements**

Gas-phase concentrations of HOMs and sulfuric acid were measured ~4 m above ground using a CI-APi-TOFMS (chemical ionization atmospheric-pressure interface time-of-flight mass spectrometer). A detailed description of the instrument can be found elsewhere (Jokinen et al., 2012; Mutzel et al., 2015). Briefly, an $^{241}$Am source was used to produce nitrate ions which were electrostatically guided into the sample flow of the inlet (length = 28 cm, inner diameter = 1.6 cm) to give nitrate clusters with gas-phase compounds present in the sampled air (sample flow ~10 L·m$^{-3}$). Then, the resulting clusters were transferred into the high vacuum region and detected by TOFMS. Calibration of the instrument was performed using sulfuric acid detection via $H_2SO_4 + (HNO_3)_nNO_3^-$ (n = 0,1,2,3) (Eisele and Tanner, 1993; Mauldin et al., 1998) with a calibration factor of $1.85 \times 10^9$ molecules cm$^{-3}$ (Berndt et al., 2014). Diffusion-controlled wall losses in the sampling tube of 12% were taken into account, using a diffusion coefficient of 0.08 cm$^2$·s$^{-1}$. The detection limit was about $10^4$ molecules cm$^{-3}$. Due to differences in ion transmission for sulfuric acid and HOMs, an uncertainty factor of 2 is estimated for the given HOM concentrations.

**2.5 T-SMPS Measurements**

Particle number size distributions were measured ~6 m above ground with a twin scanning mobility particle sizer (SMPS) custom-built by TROPOS (Leipzig, Germany) according to the design recommended by Wiedensohler et al. (2012). The instrument includes membrane dryers to keep the relative humidity below 40% both in the sample and the sheath flow. The aerosol sample is brought to bipolar charge equilibrium using a commercial $^{85}$Kr neutralizer and sent to a Hauke-type differential mobility analyzer (DMA). The mobility diameter range from 10 nm to 710 nm was scanned in 71 size bins with a time resolution of 5 min. The closed-loop sheath flow rate was set to 5 L min$^{-1}$ while the sample flow was directed to a Model 3772 condensation particle counter (TSI Inc., Shoreview, Minnesota, USA) for particle detection with a flow rate of 1 L min$^{-1}$.

## 2.6 VOC measurements

VOCs were actively sampled on commercial two-stage cartridges filled with Tenax TA/Carbograph 5TD (Markes International, Cincinnati, Ohio, USA) for 30 min with a flow rate of 0.1 L min$^{-1}$ in and above the spruce canopy at 12 m and 31 m above ground level for subsequent offline gas chromatographic analysis. Samples were taken during daytime from 09:00 to 20:00 (CET) on four selected days during F–BEACh 2014. Ozone scrubbers coated with potassium iodide were used to minimize oxidation of collected compounds. After sampling, the cartridges were sealed immediately with metal caps, placed in a screw-cap PTFE container and kept refrigerated until analysis. In the laboratory, VOCs were analyzed using standard thermal desorption gas chromatography with flame ionization detection (TD–GC–FID). The sample cartridges were thermally desorbed (200 °C), pre-focused on a Peltier-cooled trap (–15 °C), and injected onto an Rxi-5ms column (30 m, 0.32 mm, 1.00 µm, Restek, Bad Homburg, Germany) in a Sichromat 1 (Siemens AG, Germany) gas chromatograph. Monoterpenes (i.e. $\alpha$-/$\beta$-pinene, $d$-limonene, $\Delta$-3-carene, camphene) were quantified using authentic standards.

## 2.7 Filter Sample Analysis Using UHPLC–(-)ESI–HRMS

Only a brief description of the preparation and analysis of filter sample extracts will be given here. For more details, the reader is referred to the Supplemental Material.

Filter samples were taken twice a day on tetrafluorethylene-coated borosilicate filters. The sampling time was ~8 hours for daytime filters (9 a.m.–5 p.m.) and ~16 hours for nighttime filters (5 p.m.–9 a.m.). After sampling, the filters were stored at <-18 °C until analysis. For the extraction procedure, a filter sample was cut into pieces and extracted using a methanol/water solution. After sonication and evaporation to dryness, the residue was dissolved in a solution of acetonitrile/water (1:4). To compensate for losses during the processing, an average recovery rate was determined for pinic acid, which served as a surrogate for the quantification of other monoterpene oxidation products. Here, an average recovery rate of 85% was found and applied to the detected organic compounds. The LC separation was conducted on a C18 column which was coupled to a high resolution mass spectrometer (Q-Exactive, Thermo Scientific, Germany; resolving power of $R$=7·10$^4$ at $m/z$ 200). Ionization was carried out using electrospray ionization (ESI) in the negative mode. Each sample was measured in triplicate. The obtained LC–MS data were analyzed by a commercial non-target screening software (Sieve 2.2, Thermo Scientific, USA). For the elemental formula assignments, the following isotopes and conditions were used: $^{12}$C (0–50), $^{1}$H (0–100), $^{16}$O (0–40), $^{14}$N (0–4) and $^{32}$S (0–4). The mass tolerance was set to ±5 ppm. Afterwards, the obtained compound list was checked for chemically unreasonable formula assignments, such as the absence of hydrogen in carbon-containing compounds or impossible O/C ratios (O/C < 3; 0.1 < H/C < 6).

## 3 Results and Discussion

### 3.1 Detection of acidic oxidation products in SOA particles using online and offline mass spectrometry

In total, the automated non-target analysis of the filter samples by LC–MS resulted in 695 compounds which showed significant signal intensities after background subtraction. In order to identify characteristic compounds for the organic aerosol fraction from this relatively large number only those signals were selected for the subsequent analysis which showed an integrated peak area of $>10^7$ a.u. for at least two separate filter samples, resembling a signal to noise ratio of of $\geq 12$. As an additional criterion, the formula assignment for these signals had to show a mass accuracy in the range of $\pm 2$ ppm. Eventually, these thresholds led to 93 compounds in the nominal mass range of $m/z$ 133–387 which were identified from the data analysis (Fig. 1 and Supplemental Material). In general, the entire group of CHO compounds can be assigned to the class of organic acids since all measurements were carried out in the negative ion mode which is selective towards acidic compounds. Among the identified organic acids several common biogenic SOA marker compounds were detected, such as pinic acid ($m/z$ 185.0819, [M–H]$^-$) (Yasmeen et al., 2011) and terpenylic acid ($m/z$ 171.0663, [M–H]$^-$) (Claeys et al., 2009). The concentrations of all SOA marker compounds in PM$_{2.5}$ were estimated using pinic acid as calibration standard. Despite similar chemical structures, the ionization efficiencies might, however, differ to a certain extent among these marker compounds. For example, in a post-calibration experiment the response of the MS for MBTCA was found to be ~80% of the one for pinic acid. Furthermore, the composition of the LC eluent can have additional effects on the actual ionization efficiencies. Therefore, the given values should rather be taken as semi-quantitative, and in the case of MBTCA considered as a lower limit. Table 1 gives an overview of the identified marker compounds and their average concentrations during the campaign period. A comprehensive list of all detected compounds is given in the Supplemental Material.

The most dominant marker compounds during the campaign period were MBTCA and 3-carboxyheptanedioic acid with estimated concentrations of 13.8 ($\pm$ 9.0) ng·m$^{-3}$ and 10.2 ($\pm$ 6.6) ng·m$^{-3}$, respectively. While MBTCA depicts a major oxidation product of $\alpha$-/$\beta$-pinene (Szmigielski et al., 2007; Müller et al., 2012), 3-carboxyheptanedioic acid is a major oxidation product of $d$-limonene (Jaoui et al., 2006). These findings suggest that the site was strongly influenced by biogenic emissions consisting mainly of $\alpha$-/$\beta$-pinene and $d$-limonene and their corresponding oxidation products. This hypothesis is further supported by monoterpene measurements which showed relative mixing ratios of 38% $\alpha$-pinene, 23% $\beta$-pinene, 19% $d$-limonene, 12% $\Delta$-3-carene and 8% camphene. The median mixing ratios of the sum of these five monoterpenes were 0.8 ppb$_V$ above the canopy and 1.6 ppb$_V$ within the canopy. In addition, from a comparison with the MEGAN emission model (Guenther et al., 2012), the five monoterpenes $\alpha$-/$\beta$-pinene, $d$-limonene, $\Delta$-3-carene, and camphene are estimated to contribute about 80% to the total monoterpene emissions at the F–BEACh site. Beside monoterpene emissions, previous studies at the site have shown that average mixing ratios for isoprene are typically in the range of 0.27–0.50 ppb$_V$. An overview on typically VOC mixing ratios at the site can be found in Klemm et al. (2006) and others (Grabmer et al., 2006; Graus et al., 2006).

Despite these relatively high mixing ratios for monoterpenes and isoprene, Plewka et al. (2006) already observed that early-generation oxidation products of isoprene and terpenes account only for a small part of the total organic carbon content of the

particles at the site. In agreement to this work, similar concentrations in the lower ng·m$^{-3}$ range were found during the F-BEACh study for early-generation monoterpene oxidation products, e.g. pinonic acid ($c = 2.9$ ($\pm$ 2.8) ng m$^{-3}$) and pinic acid ($c = 4.7$ ($\pm$ 2.5) ng·m$^{-3}$). However, since MBTCA as well as 3-carboxyheptanedioic acid are known to be formed via photooxidation of their monoterpene precursors (Jaoui et al., 2006; Szmigielski et al., 2007; Müller et al., 2012), this observation might indicate the occurrence of fast photochemical aging processes, eventually resulting in high abundances for these compounds during the campaign period. This hypothesis is also in agreement with high solar radiation values observed at the site, typically showing a maximum around midday at an average of 393 ($\pm$ 15) W·m$^{-2}$. In addition, real-time measurements of the organic aerosol fraction and trajectory calculations also suggest a photochemical source for these compounds, as it will be discussed later on in the text. Besides several monoterpene oxidation products, also a marker compound for sesquiterpene oxidation, i.e. $\beta$-nocaryophyllinic acid, could be identified (van Eijck et al., 2013). However, the contribution of sesquiterpene oxidation products on particle composition cannot be estimated here because the observed concentrations were typically below the quantification limits.

As can be seen from Fig. 1, several sulfur and nitrogen-containing compounds were found on the filter samples, i.e. CHOS, CHON, and CHONS. Similar to the CHO group, all these compounds have to exhibit a certain acidity which allows the detection as [M–H]$^-$ ions in the negative ion mode. Therefore, the CHOS and CHONS compounds were assigned to organosulfates and nitrooxy organosulfates, respectively, which contain an acidic organic sulfate (R–OSO$_3$H) functionality. It should, however, be noted that hydroxysulfonates are isobaric with organosulfates, and thus, might contribute to a certain extent to this class. The CHON group might possibly comprise acidic organonitrates, although, no further evidence can be given here. While only 4% of the number of compounds were classified as CHON compounds about 47% of the compounds are either belonging to the CHOS or the CHONS group. This large number of organosulfates and nitrooxy organosulfates is, however, not surprising since these compound classes are ubiquitously found in organic aerosol particles and readily accessible for deprotonation via electrospray ionization (Iinuma et al., 2007; Surratt et al., 2007, 2008; Hallquist et al., 2009; Altieri et al., 2009; Schmitt-Kopplin et al., 2010; Gómez-González et al., 2012; Lin et al., 2012; Kahnt et al., 2013; O'Brien et al., 2014; Staudt et al., 2014; Nozière et al., 2015; Riva et al., 2015, 2016a,b). A comprehensive list of all sulfur- and nitrogen-containing compounds is given in the Supplemental Material.

Several of the identified sulfur-containing compounds were already studied in the past and found in field and laboratory studies (Liggio and Li, 2006; Surratt et al., 2007; Surratt et al., 2008; Altieri et al., 2009; Kristensen et al., 2011; Nguyen et al., 2012; Lin et al., 2012; Kristensen et al., 2016). In general, it is assumed that organosulfates and nitrooxy organosulfates have a mixed biogenic/anthropogenic origin, possibly involving particulate sulfuric acid, SO$_2$, NO$_X$ and radical-initiated chemistry (Surratt et al., 2008; Zhang et al., 2009; Nozière et al., 2015). As can be seen from Table 2, among the CHOS compounds several highly oxidized organosulfates (HOOS) were found on the filter samples. This recently described compound class exhibits O/C ratios greater than 1.0 and is possibly connected to the presence of gas-phase HOMs and, thus, might have implications for new particle formation processes (Ehn et al., 2014; Mutzel et al., 2015).

Real-time analysis of aerosol particles reaching the site was carried out using a HR–ToF–AMS and the recently described AeroFAPA–MS (Brüggemann et al., 2015). While the AMS was used for a general classification of the aerosol particles' components in ammonium, sulfate, nitrate, chloride and organics, the AeroFAPA–MS was resolving the organic fraction on a molecular level. In summary, the majority of the aerosol particle mass was classified as organic compounds (63.4%), followed

by sulfate (21.1%), ammonium (8.7%) and nitrate (6.7%).

Figures 2 and 3 show the concentrations of organics, measured by the AMS, in comparison to the total ion current (TIC) measured by the AeroFAPA–MS. It is assumed that the TIC, i.e. the sum of all detected ions, only shows signals for organic compounds since inorganic species are typically not volatilized and ionized by the AeroFAPA ion source. Additionally, the particle number size distributions and the main directions of 96 hours backward trajectories, calculated by HYSPLIT (HYbrid

Single-Particle Lagrangian Integrated Trajectory; Draxler and Rolph, 2013), are given for the campaign period. As can be seen from Figure 2, the signals for the 93 identified compounds by LC–HRMS and the signals of AeroFAPA–MS generally follow the same trends and are in agreement with the trend for the total organic aerosol mass, measured by the AMS. All three instruments show a maximum of signal intensities during the night of the 21$^{st}$ of July which can be explained by particles with relatively large diameters (median diameter ~150 nm) from regional sources reaching the site. This observation is further

supported by HYSPLIT backward trajectories, exhibiting rather low altitudes and trajectory lengths (Supplemental Material), as well as a strong increase in sulfate during the night (see also Fig. 5, panel b). For the organic aerosol fraction a maximum concentration of 16.9 $\mu g \cdot m^{-3}$ was determined by the AMS for this period. The days before the 21$^{st}$ of July are mainly characterized by trajectories coming from Western Europe and Northern Germany while afterwards the trajectories are arriving almost exclusively from Eastern and Northeastern Europe, i.e. Estonia and Russia.

Deviations between the signals for organics of the AMS and the total signals of the AeroFAPA–MS are mostly observed during nighttime which is possibly due to the formation of non-acidic compounds, such as alcohols, aldehydes, or ketones, possibly formed by nighttime nitrate radical chemistry, eluding detection by AeroFAPA–MS. In contrast, compounds containing organic bonded sulfate, such as organosulfates or nitrooxy organosulfates, are readily measured by the AeroFAPA–MS and LC–HRMS, whereas the AMS cannot differentiate between inorganic sulfate and organic bonded sulfate. The same bias is

typically observed for measurements of inorganic nitrate and organonitrates. Thus, all sulfate and nitrate signals of the AMS were assigned to the inorganic fraction, possibly leading to an underestimation of the organic aerosol mass (Liggio and Li, 2006; Farmer et al., 2010, Vogel et al., 2016).

In order to estimate the portion of organic compounds in aerosol particles that was measureable by AeroFAPA–MS, the signals of AMS organics and the TIC of the AeroFAPA–MS were plotted against each other. As depicted in Fig. 3 (panel a), the data

of the two instruments exhibit a linear correlation for the entire campaign period. By calculation of a linear regression fit a correlation coefficient of $R^2 = 0.83$ was determined, indicating that about 83% of the variability of the organic aerosol mass can be explained by the AeroFAPA–MS signals. Furthermore, the AeroFAPA–MS signals ([M–H]$^-$) of the 93 compounds, which were previously identified from LC–MS data as characteristic contributors to the organic aerosol fraction, were plotted as a function of the organic aerosol mass, determined by the AMS (Fig. 3, panel b). Similar to the correlation of the TIC of the

AeroFAPA–MS to the organic aerosol mass, a linear correlation was found for the 93 signals ($R^2 = 0.80$). Taking the well-established AMS as a reference and neglecting measurement uncertainties, this correlation indicates that about 80% of the organic aerosol's variability can be explained by these 93 signals, supporting the hypothesis that these compounds reflect the general behavior of the organic aerosol fraction at the site. Nonetheless, it should be noted that from the AeroFAPA–MS data

an unambiguous formula assignment is not possible due to the unit mass resolution.

Figure 4 (panel a) shows the summed AeroFAPA-MS mass spectra over the entire measurement period. As can be seen from this figure, the AeroFAPA–MS spectra support the aforementioned hypothesis that the composition of the particle phase reaching the site were influenced by BSOA marker compounds such as 2-hydroxyterpenylic acid ($m/z$ 187, [M–H]⁻), MBTCA and 3-carboxyheptanedioic acid (both $m/z$ 203, [M–H]⁻). Moreover, signals for pinonic acid ($m/z$ 183, [M–H]⁻) and pinic acid

($m/z$ 185, [M–H]⁻) remain quite low over the entire campaign period, as it was already observed for the filter samples, confirming the already mentioned low concentration of primary oxidation products at the sampling site. The ratio of signals for MBTCA and pinic acid, which can be used as aging proxy for organic aerosols (Vogel et al., 2016), shows an average value of 5.76, however, even ratios of >32 were observed for single days (Fig. S-8). Although this ratio is very specific for the instrumental setup of the AeroFAPA–MS and its ionization mechanisms, these extremely high values suggest that mainly air

masses with aged aerosol reached the site. In agreement, HYSPLIT trajectory calculations reveal that arriving air masses typically traveled several days over land with distances of >1,500 km at low altitudes, often within the boundary layer (Figures S-1 to S-6). Moreover, the majority of the trajectories are accompanied by high solar radiation without any precipitation along their way, leading to a high degree of solar radiation and, therefore, photochemical processing of the transported air masses (see Supplemental Material). Since MBTCA exhibits a rather long atmospheric lifetime of ~10 days (Nozière et al., 2015), the

observed high abundance of MBTCA might, therefore, not be solely the result of rapid photochemically-driven oxidation near the sampling site, but could be influenced to a certain extent by long range transport of organic aerosols. It should also be noted that there are two trajectories (20[th] and 21[st] of July, Figure S-1) travelling along the Czech/German border mostly over coniferous forest for at least 24 h before arriving at the Waldstein site. Previously, these southeasterly wind directions have been related to regional new particle formation events observed at the site (Held et al., 2004).

In addition to the detection of lower molecular weight BSOA marker compounds, the averaged mass spectrum exhibits several signals in the higher $m/z$-range at significant abundances. These signals might correlate to larger and more oxygenated compounds, as also observed from the LC–HRMS data. For example, AeroFAPA–MS signals at $m/z$ 357 are in agreement with the LC–MS signals for a compound at m/z 357.1559 ([M–H]⁻) with the molecular formula $C_{17}H_{26}O_8$. This compound was previously identified as a dimeric oxidation product of α-pinene (Yasmeen et al., 2010; Beck and Hoffmann, 2015). Additional

AeroFAPA–MS signals in this mass range may also correspond to the formation of sesquiterpene oxidation products as described recently by others (van Eijck et al., 2013; Chan et al., 2011; Zhao et al., ACP, 2016). As an example, signals at $m/z$ 255 might indicate the presence of β-nocaryophillinic acid ($C_{13}H_{20}O_5$), which is supported by LC–MS signals for this compound at $m/z$ 255.1238 ([M–H]⁻).

Further agreement between LC–MS and AeroFAPA–MS in the higher $m/z$-range is also observed for nitrogen-containing compounds. In general, most acidic monoterpene oxidation products show odd $m/z$ ratios in the mass spectra of the AeroFAPA–MS, since they only contain carbon, hydrogen and oxygen atoms and are detected as [M–H]$^-$ ions. However, the sum of AeroFAPA–MS spectra also exhibits elevated signals at even $m/z$ ratios, such as $m/z$ 308, showing a high linear correlation ($R^2$=0.76) to the organic aerosol mass measured by the AMS (Fig. 4, panel b). According to the nitrogen rule, these signals correspond to nitrogen-containing compounds with an odd number of nitrogen atoms. To identify this compound, the LC–MS data were checked for signals at the nominal $m/z$ ratio 308. In fact, a nitrogen-containing compound with the chemical formula $C_{11}H_{18}O_9N$ ($m/z$ 308.0987, [M–H]$^-$) was detected from the filter analysis at this nominal $m/z$ ratio. Since the AeroFAPA–MS as well as the LC–ESI–MS are selective towards acidic compounds, this signal possibly indicates the presence of a highly oxidized nitrogen-containing carboxylic acid, such as a nitrooxy carboxylic acid. Similarly, a signal at $m/z$ 250 is found in the AeroFAPA–MS spectra, showing a linear correlation to the AMS data. In this case, however, the signals of the LC–MS analysis exhibited quite low abundances. Nonetheless, one significant signal was identified at $m/z$ 250.0208, representing $C_7H_8O_9N$ ([M–H]$^-$). Due to the high oxygen content of these two compounds and their low corresponding signals from the LC–MS analysis, it is assumed that they possibly decompose during sampling, transport, storage or processing of the filter samples. Moreover, nitrooxy compounds are also known to be prone to nucleophilic substitution by $SO_4^{2-}$, forming the more stable organosulfate derivates (Darer et al., 2011). Thus, online detection methods such as AeroFAPA–MS might allow a more reliable detection of such highly oxidized nitrooxy carboxylic acids in organic aerosols. Nonetheless, different ionization efficiencies and in-source formations might also have a significant effect on the detection of such compounds and should be investigated further in the future.

## 3.2 Real-time detection of HOOS in the field

In order to investigate the presence of monoterpene-derived HOOS using real-time data of the AeroFAPA–MS, several representative compounds were chosen which were previously identified from the filter analysis (Table 2). The selection procedure for these representative compounds was based on the following criteria: Firstly, the HOOS were grouped according to their number of carbon atoms per molecule into C6 to C10 compounds. HOOS exhibiting carbon numbers <6 were discarded since they are likely to be decomposition products of larger HOOS or isoprene-derived compounds. Secondly, since the AeroFAPA–MS exhibits only unit mass resolution the signals of the representative HOOS had to show a higher intensity on their nominal $m/z$ ratio than any other signal for the LC–MS data. Moreover, special care was taken that no HOOS with identical nominal $m/z$ ratios but different carbon chain lengths were chosen as representative, as it is the case e.g. for $C_9H_{13}O_9S$ and $C_{10}H_{17}O_8S$ (both at nominal $m/z$ 297). Eventually, the high mass resolution data of the LC–MS analysis were checked for HOOS that meet these criteria.

As can be seen from Table 2 and Fig. S-7 (Supplemental Material), for each of the C7 to C10 HOOS classes one appropriate compound was found. Nonetheless, no signal of the C6 HOOS matched the selection criteria for the AeroFAPA–MS signals. Therefore, signals for HOOS containing 6 carbon atoms will not be discussed here in order to avoid wrong assignments. For

the C7 HOOS the signal at $m/z$ 239.0231 was chosen, representing $C_7H_{11}O_7S$ ([M–H]$^-$). This signal shows the highest abundance of all HOOS compounds from the filter measurements, as it was already observed by Mutzel et al. (2015), and almost no other signal was detected in significant abundances at this nominal $m/z$ ratio. For the C9 HOOS an intense signal at $m/z$ 267.0543 ($C_9H_{15}O_7S$, [M–H]$^-$) met the criteria and was selected. This finding is also in agreement with previous studies in which the C7 and C9 HOOS have been identified in laboratory and field measurements by Surratt et al. (2008). The $m/z$ ratio at 285.0284, embodying $C_8H_{13}O_9S$ ([M–H]$^-$), was chosen as representative for the C8 HOOS. Here, it should be noted that this signal was observed in lower abundances and some additional, but less distinct, signals were found at this nominal $m/z$ ratio. A similar case was observed for the C10 HOOS for which the signal at $m/z$ 327.0390 ($C_{10}H_{15}O_{10}S$, [M–H]$^-$) was chosen as representative. In general, for all of the selected compounds it was assumed that additional minor signals on the same nominal $m/z$ ratio have negligible effects on the overall signal intensity for HOOS over the entire measurement period. Furthermore, it was assumed that AeroFAPA–MS detects the selected HOOS as [M–H]$^-$ ions, and thus, at the same nominal $m/z$ ratio as for the LC–MS data. A comparison between the time traces of HOOS detected by LC–MS and AeroFAPA–MS is given in the Supplemental Material (Fig. S-10), showing similar trends for the two techniques and supporting the suitability of the selected criteria. In the following only the signals of these four representative HOOS are discussed.

Figure 5 (panel a) depicts the signals for HOOS of the AeroFAPA–MS which were plotted as a function of each other and checked for linear correlations among them. In addition, the particulate sulfate concentrations and relative humidity (RH) are given by the marker size and the color code, respectively. In general, all four HOOS classes show a linear correlation to each other, suggesting similar sources for these compounds. However, the group of C7 HOOS exhibits significant lower correlation coefficients of 0.51, 0.55 and 0.52 to the C8, C9, and C10 HOOS, respectively. This decreased correlation might indicate that the source for the C7 class is somewhat different to the larger HOOS. In fact, while the signal for $C_7H_{11}O_7S^-$ ($m/z$ 239.0231, [M–H]$^-$) shows the highest abundances of all HOOS from the filter sample analysis, as it was also reported by Mutzel et al. (2015), the AeroFAPA–MS measurements exhibit only low signals for this compound, further suggesting a different source than for the other HOOS. Possibly, this compound is a decomposition product of the larger HOOS compounds and is formed over time on the filter surface during sampling, storage and/or processing of the sample. This hypothesis is further supported by comparing time traces for HOOS on single days where the signals for the larger HOOS classes differ clearly from the signals for the C7 HOOS (Fig. S-9).

As can be seen from Figure 5, all HOOS classes yield the strongest signals for high particulate sulfate concentrations, however, no linear correlation could be observed over the entire campaign period except for single days. Moreover, the most intense signals for HOOS were observed during high RH periods which coincided with a strong increase of the particulate sulfate concentrations, following the same trend as for the HOOS signals (Fig. 5, panel b). This finding might suggest that aqueous-phase chemistry plays a major role for HOOS production, as it is known for other OS compounds (Herrmann et al., 2015). Furthermore, it should be noted that for the entire campaign period the particle acidity was very low and rather stable (average of $H^+_{Aer}$ = 7.4 nmol m$^{-3}$), indicating the presence of partially or even fully neutralized particles (Fig. S-11). In contrast to previous studies, which suggest aerosol acidity to be one of the main factors driving organosulfate formation (Surrat et al.,

2007, 2008; Iinuma et al., 2009, Gaston et al., 2014), no such effect was observed here. This result is, however, not contradicting previous findings, but rather indicating that even at low particle acidities HOOS formation can be observed, as it will be discussed in the following.

During the high RH period from $21^{st}$–$23^{rd}$ of July, the sulfate concentrations and signals for HOOS show a linear correlation ($R^2$=0.70). Figure 5 (panel b) depicts the time trace of the signals for the HOOS and the particulate sulfate concentrations. Additionally, the RH is given by the color code. Within the first hours of July $21^{st}$ the signals for HOOS and the sulfate concentrations still show some minor deviation, however, starting roughly from 9 a.m. both time traces follow almost exactly the same trend for the rest of this period, for which RH values mostly exceed 80%. During such high RH periods dissolved $HSO_4^-$ might react with HOMs after phase transition via a nucleophilic attack to give HOOS, as it was already proposed previously (Mutzel et al., 2015). An additional or alternative pathway for the formation of HOOS might be the hydrolysis of hydroperoxide-containing HOMs as recently reported for methylglyoxal-, isoprene-, and alkane-derived hydroperoxides (Lim and Turpin, 2015; Riva et al., 2016a, 2016b). Assuming that RH is a one of the main factors driving the uptake of gas-phase species into the particle phase (Shiraiwa et al., 2011, 2013), the required rapid phase transition of gas-phase HOMs is further supported by the observed trend for the sum of HOMs, measured by the CI-APi-TOFMS. In contrast to the diurnal behavior observed for other days of the campaign, the HOM concentrations show very low concentrations during this high humidity period (Fig. S-12). As depicted in the figure, until ~11 a.m. of July $22^{nd}$ the RH values are rather high (RH > 60%). During this period, i.e. ~6 p.m. of $21^{st}$ to ~11 a.m. of the $22^{nd}$ of July, the signal for gas-phase HOMs is hardly correlated to neither the HOOS signal nor the sulfate concentration and shows rather low abundances. However, as soon as the RH decreases to values below 60%, the gas-phase concentration of HOMs exhibits an immediate and strong increase, roughly tripling the sum of HOMs within ~1.5 hours. It should be noted that some strong precipitation around midnight of the $21^{st}$ of July led to the observed strong decrease in particulate sulfate as well as HOOS concentrations.

In order to further investigate the formation of HOOS and the role of possible precursor HOMs, single $m/z$ ratios of the CI-APi-TOFMS were analyzed in more detail for the $17^{th}$ and $24^{th}$ of July. Since during the high RH periods the gas-phase concentrations of HOMs were extremely low due to a rapid phase transition (Figure 6, panel b), only these two days with low RH values were chosen for a further data analysis and discussion. In general, during dryer periods, signals with an odd $m/z$ ratio in the mass spectra of the CI-APi-TOFMS dominate the sum of gas-phase HOMs (Fig. S-12). As it was reported previously (Ehn et al., 2014; Jokinen et al., 2014), several of these compounds represent peroxyradicals ($RO_2^{\bullet}$) which may act as precursors for closed-shell HOMs. Remarkably, the signals for three of the four identified $RO_2^{\bullet}$ (i.e. $C_{10}H_{15}O_6^{\bullet}$, $C_{10}H_{17}O_7^{\bullet}$, $C_{10}H_{15}O_8^{\bullet}$) and HOOS follow the same trends during the dry periods, possibly revealing a connection between these species. Solely the signals for $C_{10}H_{15}O_{10}^{\bullet}$ (i.e. $m/z$ 357, $[M+NO_3]^-$) exhibit a different behavior (Fig. S-13). As an example, Figure 6 (panel a) depicts the time traces of a C10 HOOS ($m/z$ 327, $C_{10}H_{15}O_{10}S$, $[M–H]^-$) and the most abundant $RO_2^{\bullet}$, i.e. $C_{10}H_{15}O_8^{\bullet}$ ($m/z$ 325, $[M+NO_3]^-$), which might represent a possible precursor species for this HOOS compound. In addition, the gas-phase concentration of $H_2SO_4$ (divided by 4) and RH are given. As can be seen for July $17^{th}$ and $24^{th}$ the signal for $RO_2^{\bullet}$ is increasing with time, showing its maximum at ~$4 \cdot 10^6$ molecules $cm^{-3}$ for both days around 11 a.m., and is afterwards slowly decreasing

again. In each case, the signal for the C10 HOOS follows the concentration of the RO$_2^\bullet$. While the observed coinciding concentration profiles are not unambiguous for the limited available dataset, there might be a certain connection between RO$_2^\bullet$ and the observed HOOS. In contrast, time series for a possible closed-shell HOM precursors, such as C$_{10}$H$_{16}$O$_{10}$ ($m/z$ 358, [M+NO$_3$]$^-$), show only a weak agreement with signals for HOOS (Fig. S-14). As previously suggested, e.g. by Kurtén and co-workers (2015), RO$_2^\bullet$ contain acylperoxy-functionalities which might possibly undergo a nucleophilic attack by HSO$_4^-$, forming the corresponding HOOS, which has been discussed for closed-shell HOMs earlier by Mutzel et al. (2015). Such a mechanism would explain HOOS formation coupling to RO$_2^\bullet$ in the particle-phase and/or at the interface. However, knowledge on the existence of such formation pathways still needs to be much better explored.

For both days also a significant amount of gas-phase H$_2$SO$_4$ was present, showing maximum values of ~1.6·10$^7$ molecules cm$^{-3}$ and ~1.3·10$^7$ molecules cm$^{-3}$ for July 17$^{th}$ and 24$^{th}$, respectively. It should be noted that in contrast the particulate sulfate concentration was rather high during July 17$^{th}$ (maximum at ~5.7 µg m$^{-3}$) but quite low for July 24$^{th}$ (maximum at ~2.8 µg m$^{-3}$). In principle, gas-phase H$_2$SO$_4$, which gets rapidly dissolved in the aqueous phase during high RH periods, might present an additional sulfur source for HOOS generation during such dryer periods. However, correlations during the campaign period were only observed for single days when high HOOS signals coincided with elevated gas-phase H$_2$SO$_4$ concentrations (Fig. 6, panel b). Thus, due to the limited data set and ambient conditions the exact reaction mechanisms for HOMs and HOOS cannot be discriminated here. In general, it can also be expected that the presence of precursor HOMs is the rate-limiting step in HOOS production, since concentrations for particulate sulfate as well as gas-phase H$_2$SO$_4$ are typically significantly higher. The afore-mentioned reaction mechanisms for the high RH periods, i.e. nucleophilic attack by HSO$_4^-$ might, therefore, still play an important role for the dryer periods.

## 4 Conclusions

In this study complementary mass spectrometric techniques were used for the analysis of the ambient organic aerosol fraction during the F–BEACh 2014 field campaign in Central Europe. A non-target analysis of filter samples by LC–HRMS showed elevated concentrations for 93 acidic oxygenated hydrocarbons, which were, therefore, assigned as characteristic contributors to the organic aerosol mass at the site. VOC measurements indicated high mixing ratios for monoterpenes, such as $\alpha$-/$\beta$-pinene. However, especially later-generation monoterpene oxidation products were observed in higher concentrations in the particle phase, suggesting a rapid oxidation of these precursors. In addition, long-range transport of aged air masses was possibly influencing aerosol chemistry during the campaign period, eventually leading to elevated concentrations for aging markers such as MBTCA. In particular, the comparison of concentrations for early-generation and later-generation oxidation products, such as pinic acid and MBTCA, indicates that photochemically aged aerosol masses were present on several days. HYSPLIT trajectory calculations further supported this hypothesis by giving large trajectory lengths (>1,500 km) for arriving air masses, traveling over land under high solar irradiation. Additionally, around 47% of the tentatively identified compounds were sulfur-

containing, suggesting a rather high anthropogenic impact on biogenic emissions and their oxidation processes. Among these sulfur-containing compounds, several OS, NOS, and HOOS were detected.

Real-time measurements of the aerosol constituents using AeroFAPA–MS, AMS and CI-APi-TOFMS further supported these findings and correlations among HOOS classes, sulfate and gas-phase HOMs were investigated. In agreement with previous studies the results support the assumption that monoterpene-derived HOOS are formed by reactions of gas-phase HOMs with particulate sulfate, i.e. $HSO_4^-$ (Mutzel et al., 2015). Nonetheless, since signals for the C7 HOOS showed rather low abundances from the real-time data and only low correlations to other HOOS, it is assumed that these smaller HOOS might represent decomposition products of larger HOOS. This finding is, however, not contradicting previous publications which found the highest concentrations for C7 HOOS from the analysis of filter samples (Mutzel et al., 2015) but rather suggesting that larger HOOS decompose not only in the atmosphere but also during filter sampling, storage or processing. Furthermore, high RH periods indicated that aqueous-phase chemistry is presumably also playing a major role in HOOS production, since the highest HOOS signals coincided with high RH values and high particulate sulfate concentrations. Interestingly, no correlation between particle acidity and HOOS formation was observed here, as it was reported for less-oxygenated OS (Surratt et al. 2007, 2008; Iinuma et al., 2009, Gaston et al., 2014). For dryer periods, gas-phase $RO_2^\bullet$ might serve as additional direct or indirect precursors for HOOS, however, further evidence is needed here.

*Acknowledgements*. M.B. and T.H. thank the Max Planck Graduate Center with the Johannes Gutenberg-Universität Mainz (MPGC) for financial support. The authors gratefully acknowledge the NOAA Air Resources Laboratory (ARL) for the provision of the HYSPLIT transport and dispersion model and READY website (*http://www.ready.noaa.gov*) used in this publication. Moreover, the authors gratefully acknowledge the Department of Micrometeorology at the University of Bayreuth for providing the RH data.

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

**Table 1.** Common BSOA marker compounds which were identified among the CHO compounds. The average concentrations were determined using pinic acid as reference. Standard deviations are given in brackets. A complete list of all identified CHO compounds can be found in the Supplemental Material.

| formula for [M–H]⁻ | measured *m/z* | assigned compound | average conc. / ng m⁻³ |
|---|---|---|---|
| $C_8H_{11}O_4$ | 171.0663 | terpenylic acid | 6.4 ($\pm$ 3.8) |
| $C_8H_{11}O_5$ | 187.0612 | 2-hydroxyterpenylic acid | 7.7 ($\pm$ 5.0) |
| $C_8H_{11}O_6$ [a] | 203.0561 | MBTCA | 13.8 ($\pm$ 9.0) |
| | | 3-carboxyheptanedioic acid | 10.2 ($\pm$ 6.6) |
| $C_9H_{13}O_4$ | 185.0819 | pinic acid | 4.7 ($\pm$ 2.5) |
| $C_{10}H_{15}O_3$ | 183.1027 | pinonic acid | 2.9 ($\pm$ 2.8) |
| $C_{10}H_{15}O_6$ | 231.0874 | diaterpenylic acid acetate | 5.2 ($\pm$ 2.7) |

[a] isobaric compounds

**Table 2.** Identified highly oxidized organosulfates (HOOS) by LC–MS from filter sample extracts. Compounds that were selected for further analysis from real-time data are marked with an asterisk. A comprehensive list of all detected sulfur- and nitrogen-containing compounds is given in the Supplemental Material.

| formula for [M–H]⁻ | measured $m/z$ | $\Delta m$ / ppm | O:C |
|---|---|---|---|
| $C_7H_{11}O_7S$ * | 239.0231 | 0.0 | 1.0 |
| $C_7H_{13}O_7S$ | 241.0385 | -1.0 | 1.0 |
| $C_7H_7O_8S$ | 250.9868 | 0.3 | 1.1 |
| $C_7H_9O_8S$ | 253.0028 | 1.7 | 1.1 |
| $C_8H_{11}O_9S$ | 283.0127 | -0.8 | 1.1 |
| $C_8H_{13}O_9S$ * | 285.0284 | -0.6 | 1.1 |
| $C_8H_{13}O_{10}S$ | 301.0231 | -1.3 | 1.3 |
| $C_9H_{15}O_7S$ * | 267.0543 | 0.0 | 0.8 |
| $C_9H_{13}O_8S$ | 281.0334 | -0.9 | 0.9 |
| $C_9H_{13}O_9S$ | 297.0282 | -1.3 | 1.0 |
| $C_{10}H_{15}O_{10}S$ * | 327.0387 | -1.4 | 1.0 |
| $C_{10}H_{13}O_{11}S$ | 341.0183 | -0.3 | 1.1 |

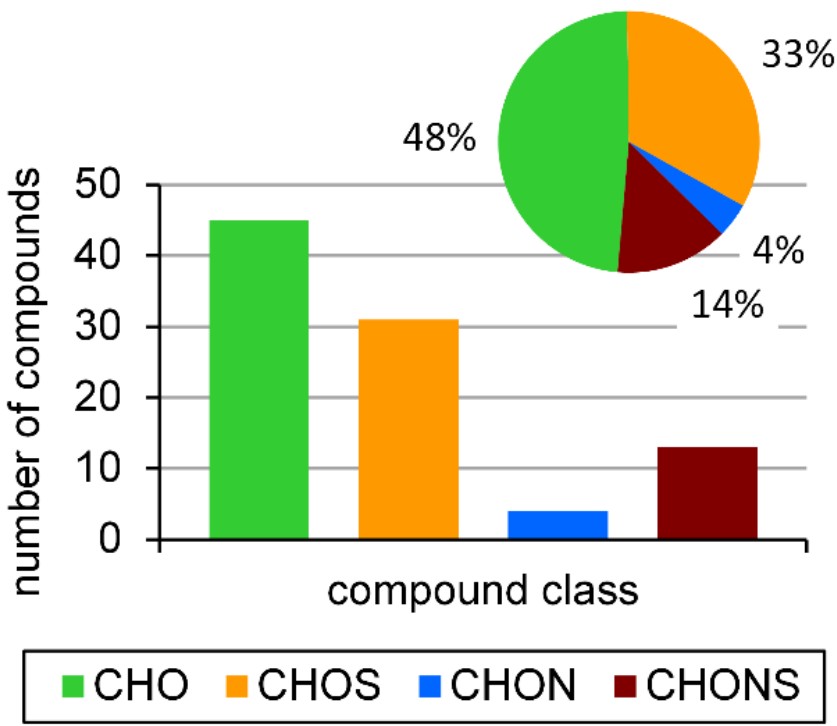

**Figure 1.** Number and fraction of identified compounds by LC–MS analysis of filter sample extracts for each compound class.

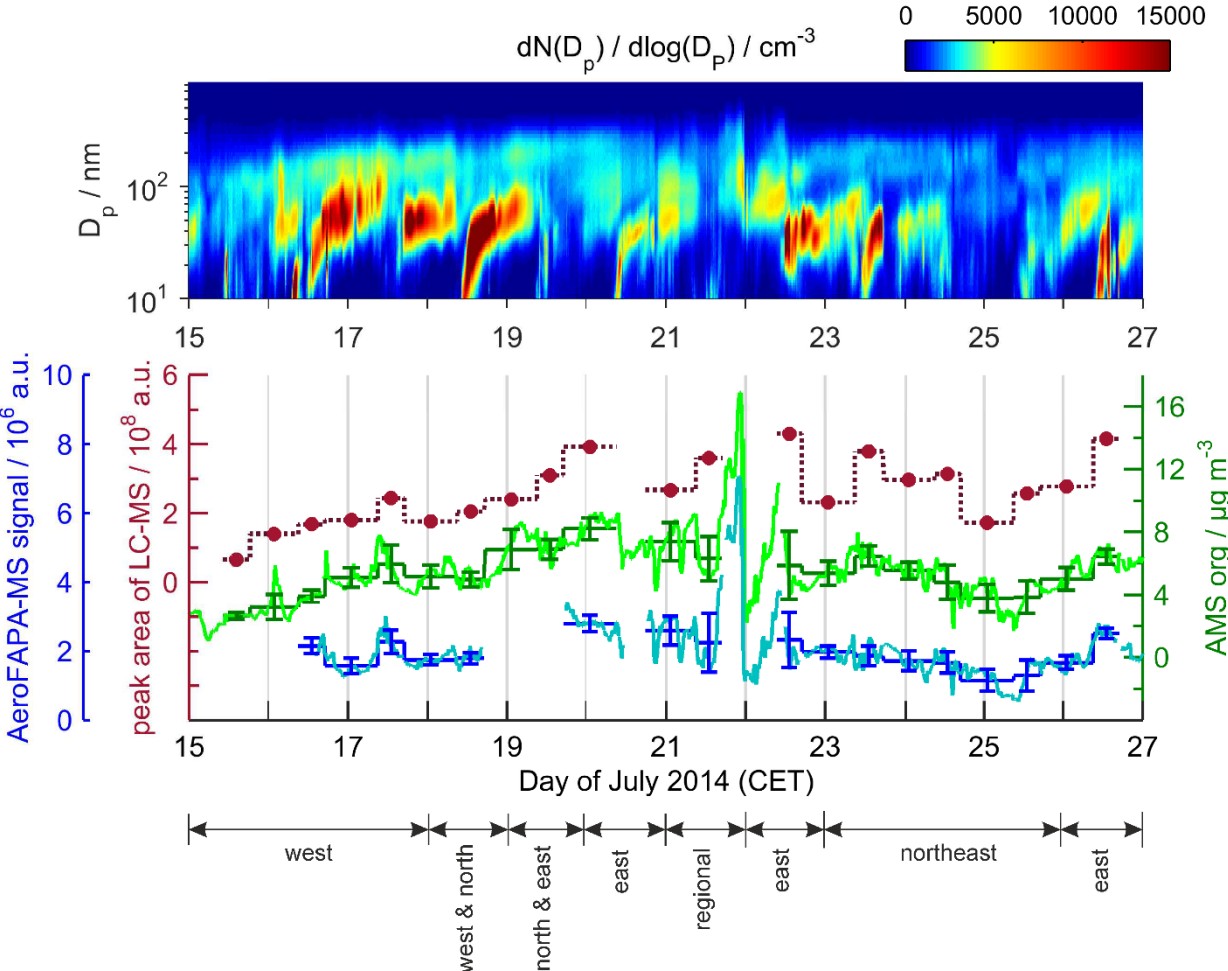

**Figure 2.** Top panel: Number size distribution of aerosol particles which was measured by the SMPS. Middle panel: Sum of the peak areas for the 93 identified compounds from the filter samples by LC–HRMS (red). The signals for these compounds, measured by AeroFAPA–MS (blue), and the organic aerosol mass, measured by an AMS (green), show similar trends. Averaged values for the filter sampling times are depicted by the horizontal lines (errorbars show one standard deviation). Bottom panel: Major source directions of 96 hours backward trajectories arriving at the site (250 m above ground level).

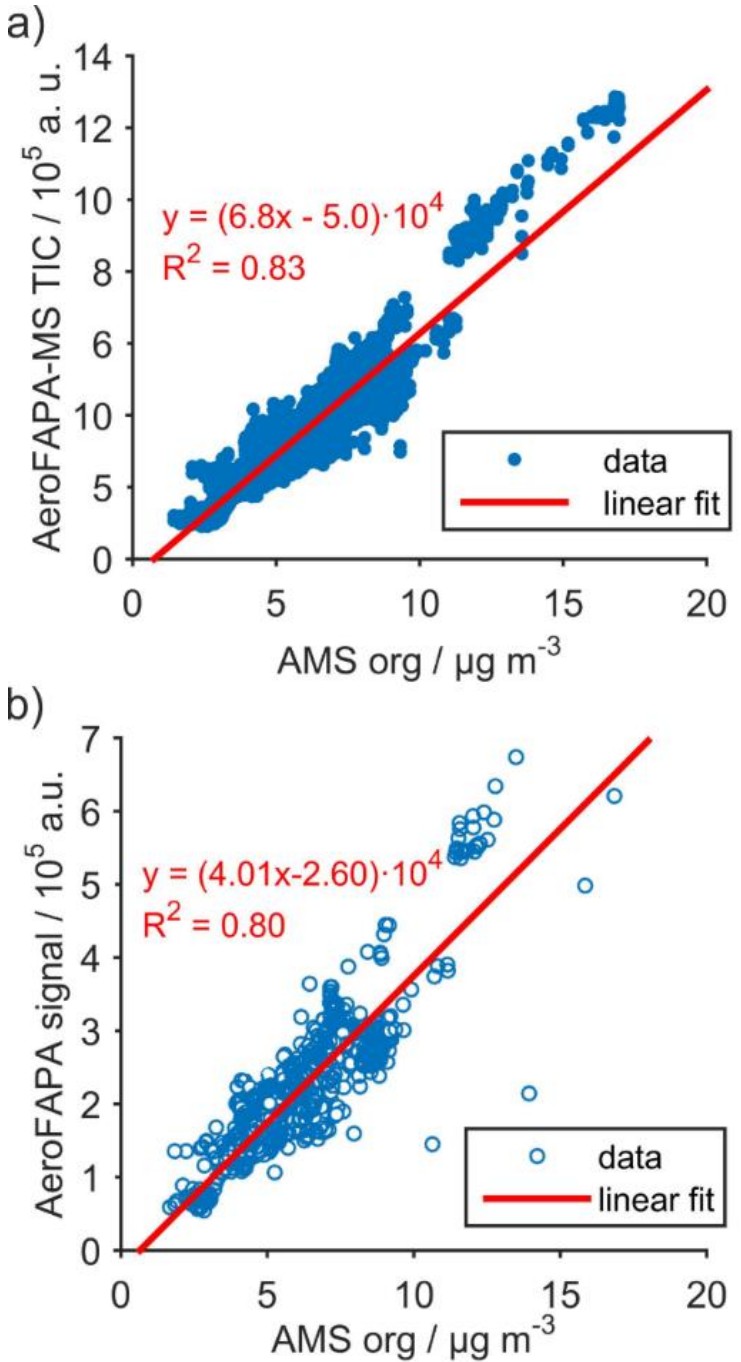

**Figure 3.** Correlation between organic aerosol mass (AMS org) and AeroFAPA–MS signals. (a) Total ion current of AeroFAPA as function of organic aerosol mass (blue dots) and linear fit (red line). (b) AeroFAPA–MS signals for compounds, that were identified by LC–MS analysis of filter samples, as a function of organic aerosol mass (blue circles) and linear fit (red line).

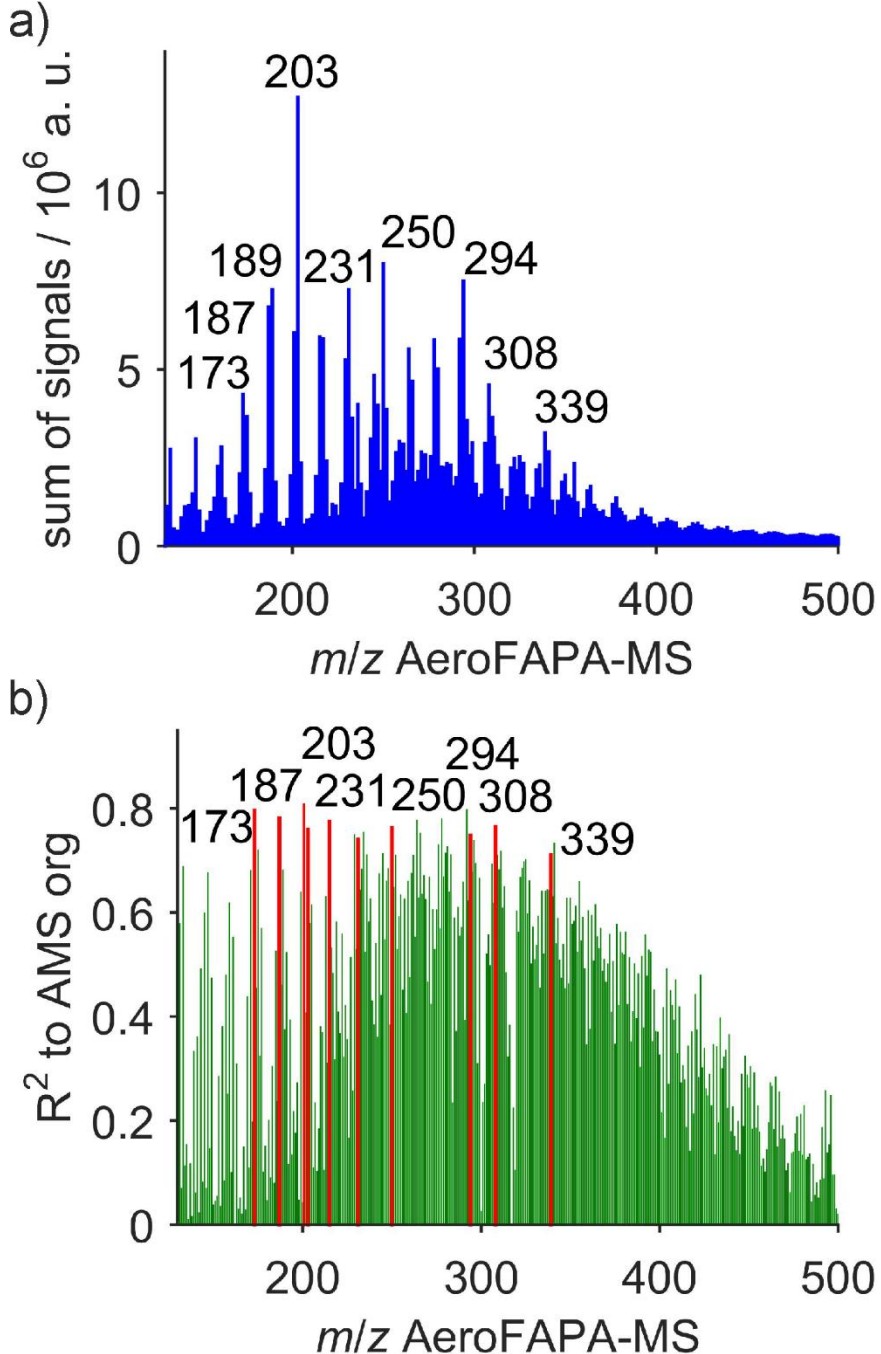

**Figure 4.** (a) Sum of signals during the campaign period of the AeroFAPA–MS. (b) Linear correlations between *m/z* ratios of AeroFAPA–MS and total organic aerosol mass measured by the AMS.

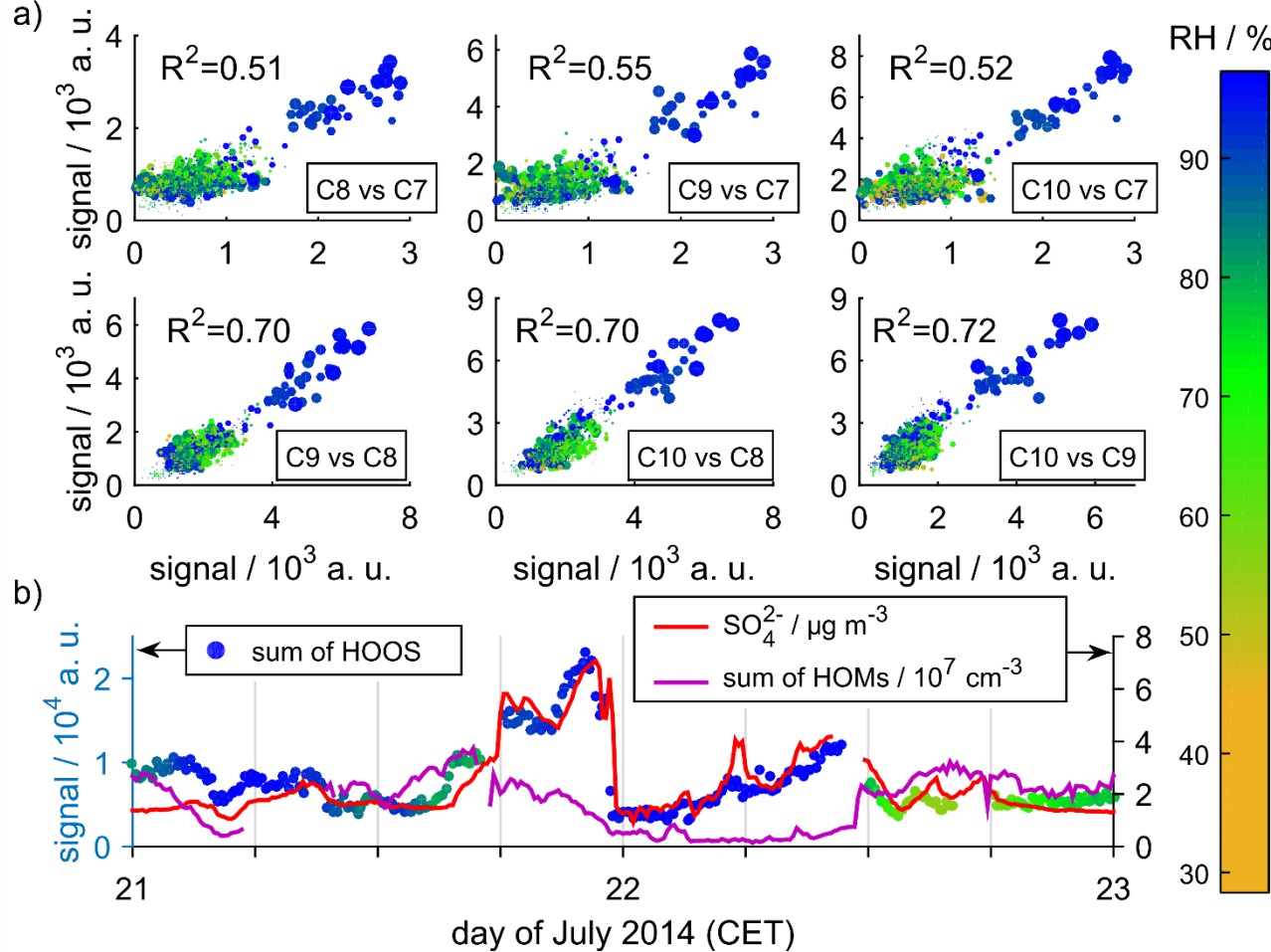

**Figure 5.** (a) Correlations among the selected HOOS signals as well as the effect of RH (color code) and particulate sulfate on their abundance (marker size, range: 0.8–7.2 µg·m$^{-3}$). (b) Comparison of the sum of HOOS signals, particulate sulfate, and sum concentration of gas-phase HOMs during July 21$^{st}$–23$^{rd}$, demonstrating good agreement between HOOS and sulfate for high RH periods.

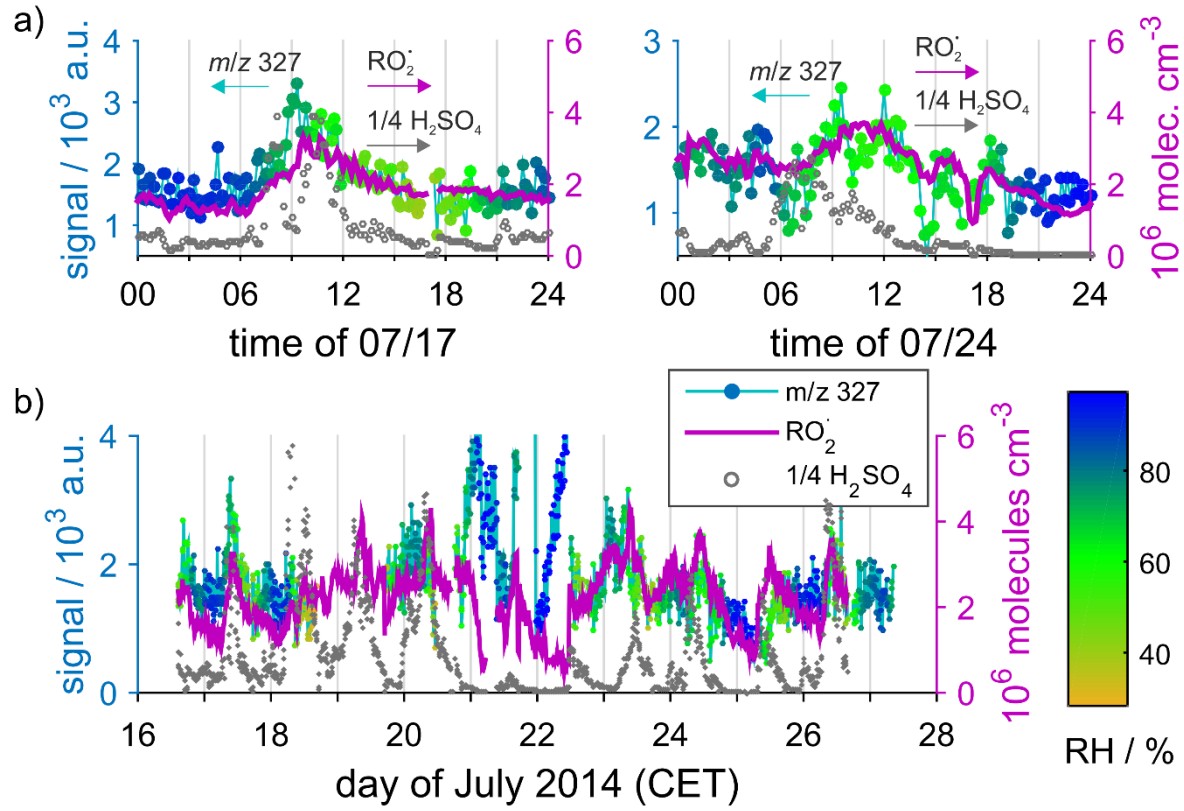

**Figure 6.** Signals for C10 HOOS (*m/z* 327), gas-phase $H_2SO_4$, $RO_2^•$ ($C_{10}H_{15}O_8^•$), and RH. (a) Time traces for the signals for July 17th and 24th, showing good agreement between C10 HOOS and $RO_2^•$. (b) Time traces for the entire campaign period, demonstrating the influence of RH on HOOS formation and $RO_2^•$ abundance. For better visibility the HOOS signal is allowed to go off scale for July 21st and 22nd.