# Peer review of "Real-time detection of highly oxidized organosulfates and BSOA marker compounds during the F–BEACh 2014 field study"

_Atmospheric Chemistry and Physics, 2016_

## Referee Comment (RC1) · Anonymous Referee #1 · 5 Sep 2016

Brüggemann et al. describe a detailed mass spectrometric analysis of organic aerosols during a field study in a forest in southeast Germany in July 2014. The array and novelty of the gas- and particle-phase analyses completed were impressive. The opportunity to compare trends in HOMs, sulfate, and HOOS is exciting and has the potential to improve our knowledge of organosulfate chemistry. However, much of the discussion is general with more focus on the methods and comparison of those methods rather than the science. I suggest major revisions with focus the results pertaining to advancing our knowledge of biogenic SOA. In particular, the detailed comparison of specific HOMs and HOOS in section 3.2 is exciting and should be expanded and examined in greater detail with additional ion markers examined and compared to the LC-HRMS results.

Additional discussion of the methods and comparison of the methods are suggested to be moved to the supplemental.

The abstract can be improved significantly. Much of the abstract reads as a list of methods and types of compounds measured with only vague, general aspects of the results stated, such that the unique findings and implications are not as clear as possible. It would be improved by focusing more on the main details of the scientific results and implications of the study, rather than the analytical aspect of the study. Similar advice should be considered for the conclusions section.

The discussion in section 3.1 is general within minimal in-depth analysis and much focus on the methods, rather than the results. The statement on lines 21-24 (page 7) oversimplifies the pathway from emission to SOA formation and does not account for oxidation pathways, reaction kinetics, and reaction yields. Then on line 28 it is stated that "rather low abundances were found for early-generation oxidation products". What is considered "rather low" compared to expected based on other studies? Do the volatilities of the expected compounds suggest that they should be significantly in the particle phase? On line 31 (page 7), it is stated that "…fast photochemical aging processes, eventually result[ed] in high abundances…"; without knowledge of the oxidant concentrations or reference to solar radiation measurements, is difficult to assess this general statement. The discussion of organosulfates and related compounds (lines 3-11, page 8) is general, focusing most on the method. However, many papers have been published using UPLC-HR-MS to study organosulfates, so this level of detail about the method is not necessary to include in the main text. I would suggest moving elsewhere (supplemental?) and combining this and the following paragraph. Additionally, the list of previous papers on line 14 (page 8) is not comprehensive as currently implied. This section suffers most from a lack of discussion of the properties of the HOOS, which appears to be a focus of the paper. Again, the text on lines 21-24 and 27-28, for example, focuses on the methods, rather than the scientific results. Other examples can be provided. Overall, the authors might consider reorganization of section 3.1 to first

discuss MEGAN results, then VOC results, then met/HYSPLIT information, and finally SOA results to provide justification first for the observed biogenic SOA markers. The case study of July 21 seems useful, and it might be helpful to also include LC-HRMS individual compound results in this examination for greater interpretation. Overall, the authors should focus on the scientific results rather than explanation of the methods, information about which should be in the methods and supplemental. Similar issues are present in section 3.2 and should be considered.

Are the authors suggesting that the RO2 directly reacts with sulfate in an implied heterogeneous reaction (Page 12, lines 30)? Why does the RO2 need to be the immediate precursor? This would contradict work of Surratt and most others in the consideration of the formation of organosulfates. However, as summarized by Xu et al 2015 (PNAS) in Fig 3, RO2 can be a precursor of organosulfates through the subsequent formation of an epoxide, followed by nucleophilic addition.

Specific Comments: Page 2, lines 2-3: It is unclear what "a good agreement" means here.

Page 2, lines 4-5: What aspect of HOOS is implied to be increased here?

Page 2, line 11: Is biomass burning SOA really considered anthropogenic? Wildfire smoke is not necessarily human-caused.

Page 2, line 8: The phrasing "Yet, until now a comprehensive structural elucidation of HOMs was not possible" implies that the structures, rather than just elemental formulas, of HOMs were determined in this study; this seems misleading.

Page 5, lines 2-3: The statement that PM1 mass closure was achieved is not necessary and is misleading, as it suggests that refractory material did not at all contribute to the aerosol at this site, which is difficult to believe.

Section 2.1: What is the forest canopy height at the field site, and what height does the canopy start at? This is needed to understand whether various measurements were

located within, below, or above the forest canopy. Also, please provide measurement heights in Sections 2.4 and 2.5. It would be helpful if an overview of measurement locations could be provided in Section 2.1.

Section 2.4: Please provide details regarding the sampling inlet. What are the transmission losses of the HOMs and sulfuric acid through the inlet? What were the uncertainties and limits of detection for sulfuric acid during this study?

Section 2.6: What VOCs were able to be investigated in this configuration?

Page 9, line 17: What fraction of the total OA mass did the 93 compounds comprise?

Page 9, lines 29-30: Since both MBTCA and pinic acid were quantified by LC-HR-MS, why can't these concentrations be used to calculate the "aging" ratio, rather than the raw AeroFAPA-MS signals?

Page 9, line 33: By "low altitudes", do you mean within the boundary layer?

Page 11, lines 30-32: The authors should consider the work of Xu et al 2015 (PNAS) who examined in detail correlations between organosulfates, RH, and sulfate concentrations. This may improve interpretation of these results.

Figure 4: The results of this figure should be discussed in greater detail in the text, taking the LC-HRMS data into greater consideration.

Figure 5: This figure could be moved to the supplemental, as the main scientific point is not clear.

Technical Comments: Page 3, lines 7-8: Fix reference location.

Page 4, Section 2.2: The first two sentences should be combined with the following paragraph to be one full paragraph.

Section 2.6: The two paragraphs in this section can be combined to avoid the presence of a two sentence paragraph.

Page 6, line 11: Why is the ratio listed as 2:8 instead of 1:4?

Page 6, lines 16-17: The phrasing of this sentence does not make sense.

Page 6, line 24: Given the mass resolving power and tolerance (given on lines 21 and 31), for what m/z range can compounds be unambiguously determined?

Page 6, line 29: The "signal threshold. . .for the detection of significant signals" is stated and in arbitrary units. It would be more meaningful if a S/N threshold were listed.

Page 7, lines 4-6: Similar to the previous comment, there is no context given for "integrated peak areas of >10ˆ7 a.u.". Some relationship to S/N thresholds would be useful here.

Page 7, lines 6-7: Does this mean that the previously mentioned 695 compounds (on line 3) were not within a mass accuracy range of +/- 5 ppm? This would be concerning. In addition, it would be useful to provide the m/z range of the identified compounds.

Page 7, line 11: I am guessing that the authors mean "and" instead of "or" here.

Page 7, lines 11-15: This calibration information could be moved to the methods or supplemental so that the authors can focus on science here. Also, it is not clear here whether all 93 compounds were calibrated with pinic acid or just pinic and terpenylic acid.

Table 1: Why are m/z 255 and 357 listed here if they were below the LOQ? Was there any day/night difference observed?

Page 7, lines 19-21: This suggests that MBTCA is a minor (in terms of yield) pinene oxidation product. Is this correct? Please clarify the comparison.

Page 7, lines 23-24: Were these the only VOCs quantified? This information is not provided in the methods, so it is difficult to fully interpret this result.

Page 8, lines 1-2: Move this statement to the methods section.

Supplemental, Page 1: For clarity, please add a header with the title and author names to the supplement.

Fig S-7, bottom panel caption: For clarity, please indicate the instrument used.

Fig S-8, caption: Additional information for interpretation of the figure would be helpful.

Fig S-9: A legend would be helpful.

---

## Referee Comment (RC2) · Anonymous Referee #2 · 6 Sep 2016

The manuscript presents interesting new work investigating the composition of organic aerosols using a suite of state-of-the-art mass spectrometric techniques. The results are interesting, but some interpretations are not fully supported by the data in the present version of the manuscript. The development of new instrumentation to investigate organic aerosol composition is important and exciting, but it is necessary to have a deep respect for the possible new artefacts and biases associated with the techniques.

Specific comments Check that the word "identification" is used correctly (according to Nozière et al., 2015) throughout the manuscript.

Abstract The abstract needs a thorough revision to correctly reflect the findings. Page 1 Line 20: How do you "identify a characteristic contributor"? L23-24: I would not

characterize concentrations around 10ng/m3 as "high". Please be specific instead. L24. The present data does not support the statement that terpene oxidation products dominate the organic aerosol fraction. The AMS data shows OA mass in the range 2-8 microgram/m3, while only few terpene oxidation products have been identified and quantified with a total concentration of about 1% of this mass.

L25. How high? L28: Did the air masses pass areas of high BVOC emissions such as coniferous forests? L32: How do you define "unambigious identification"? Generally authentic standards would be needed to support this.

P2 L4: The word "reveal" should generally be avoided in the scientific literature. Here you could use "indicate" instead.

P3 L4: Why is important to state that the LC-MS analysis is non-target, since most such analyses would be this? I suggest removing the term throughout the manuscript.

P4L11: Were there any size-selection? A previous paper (Brüggemann et al., EST 2015) describes an SMPS system in front of the AeroFAPA-MS. P4L3: Are all organic components evaporated at 200C? Will organosulfates evaporate at 200C? What happens to components remaining in the particle phase including inorganic species? The answers to these questions must be given in the manuscript.

P5L1: "data quality insurance" - do you mean quality control of the data?

P6L7: The extraction solvent seems relatively polar. Was the extraction efficiency of larger carboxylic acids (such as pinic acid) and organosulfates investigated? Could differences in extraction efficiency have affected the relative proportion between e.g. C7 and C10 organosulfates? P6L11-12: Is it correctly understood that the average recovery only reflects the loss during evaporation, not extraction efficiency?

P7L1- and Table 1: It is impressive that 695 individual compounds are eluted in only 4.1 minutes (according to information on the UHPLC gradient). How were possible matrix effects (leading to signal suppression during ESI) avoided or corrected? Why

was pinonic acid not quantified? This would have been useful for the discussion of photochemical aging. The higher concentration of MBTCA compared to e.g. pinic acid could indicate that the BSOA components are long range transported rather than locally produced. It would be useful to compare the concentrations in Table 1 to previous measurements in similar areas. Please remove compounds below limit of detection from Table 1 and just mention them in the text.

P8L11: It is a bit surprising that Hallquist et al. already in 2009 made such general conclusions on the ubiquity of organosulfates and nitrooxy organosulfates, given the very few studies conducted at that time. Please recheck or update the reference.

P9L7: Have any of the commercially available standards for organosulfates been analyzed with AeroFAPA-MS? Since the ionization technique is known to form adducts (Brüggemann, Karu and Hoffmann, J. Mass Spectrometry 2016), how was it investigated that organosulfates or organonitrates are not formed during analysis of complex samples such as aerosols containing both organic and inorganic components?

P9L12: What do you mean by "quantify"? P9L13-26: Just because the two data sets correlate, it does not mean that the Aero-FAPA-MS signal explains the variability in the AMS organic matter data. There could be other underlying common factors involved, such as long-range transport or photochemical processes. Without any quantification of the Aero-FAPA-MS measurements, the statements that "these compounds reflect major sources" and "particle phase was dominated by BSOA markers" remains not fully documented by the data. Please correct the sentence to reflect your findings more accurately. P9L27: Concentrations of pinonic acid were not listed. P9L28-30: Since MBTCA is an oxidation product of pinonic acid (with OH) the ratio MBTCA to pinonic acid would make more sense. Please include relevant references for using this ratio.

P10L1: "transported aerosol masses" -> "transported air masses" The text on this page is quite "lengthy" and could be shortened and clarified. P10L19-24: The extraction efficiency of larger compounds with the polar solvent could also affect the ratio. Since authentic standards of highly oxidized nitrooxy carboxylic acids have probably never been measured by Aero-FAPA-MS, the statement about "reliable detection" seems speculative. Furthermore possible in-source formation should be investigated. On-line methods are certainly important for measurements of these compounds, but further work is also needed.

P11L3 and Figure 5. The purpose of Figure 5 is not clear. It seems that one compound was not "found" but rather "selected". The figure should be moved to supplementary information. The first paragraph of page 11 could be shortened and focused. P11L7: In agreement with what? P11L17-28: Are the HOOS presented and discussed here only the compounds selected in the paragraph above? Please make this more clear - also in the text of Fig. 6. Could the difference between the HOOS classes observed with LC-MS and Aero-FAPA-MS also be due to differences in ionization efficiency and the question of extraction efficiency discussed above? P11L32-35: Higher concentration of sulfate (and pH) also affects surface uptake and reactions.

P12L8-13 and L17: It is not clear how the hypothesis of "rapid phase transition" of HOMs is supported by the present data. RH is closely related to temperature, and thus time of day, which also affects emissions of BVOC. There could thus be other explanations than condensation for the variation in level of HOMs.

It is very interesting how the levels of HOOS and peroxy radicals vary together.

P12L28-31: Please clarify how this relates to previously proposed mechanisms for OS formation.

Conclusion: Please adjust according to your answers to questions stated in this review.

Figure 2 lower panel: The figure is too overloaded with information in overlying graphs. Please make the figure more clear by e.g. moving the data sets further apart.

Figure 5 should be moved to supplementary information.

Figure 6. The figure is too busy. Part A could be moved to supplementary information. The marker size is too large for high concentrations of sulfate, which gives a bias in the understanding of the number of data points. Write e.g. C7 vs. C8 to make the figures easier to understand. Are the HOOS all compounds or just the ones selected to represent each group?

[Figure]

---

## Referee Comment (RC3) · Anonymous Referee #3 · 7 Sep 2016

The manuscript presents interesting investigation on the chemical characterization of organic aerosols collected in southeast Germany. Large variety of mass spectrometry techniques (online and offline) was deployed and the results proposed in this study could provide new insights in the organosulfate chemistry. However, the discussion/interpretation of the results are not well supported and most of the discussion is focus on the comparison of the methods. I suggest major revisions of the manuscript to better discuss the different findings of this study. In particular, the discussion on the potential formation of organosulfates from the heterogeneous oxidation of RO2 radicals is not well sustained and requires deeper investigation. For instance, only one RO2 radical was considered in the discussion (while 4 were identified) and the authors

did not consider the other pathways recently reported in the literature (i.e. hetero-geneous reactivity of organic hydroperoxides). In general, the authors should better compare/discuss their results with the existing literature.

Please revise the abstract to highlight the findings of this study

Page 2: lines 27-32: The authors mentioned the different pathways leading to the organosulfate formation but some reaction pathways are missing. Indeed, organosul-fates could be formed by either nucleophilic substitution of an organic nitrate group by sulfate (Darer et al., 2011; Hu et al., 2011), or by heterogeneous chemistry of gas-phase organic hydroperoxides, which might undergo acid-catalyzed perhydrolysis fol-lowed by reaction with sulfate ions (Riva et al., 2016a; 2016b).

In addition, the authors should also include the following references: - Reactive uptake of epoxides: Shalamzari et al., 2014; 2016 - Sulfate radicals: Schindelka et al., 2013. - SO2: Passananti et al., 2016

Page 3: lines 1-5. It has been recently reported that HOM monomers formed from the oxidation of $\alpha$-pinene are unlikely to be ELVOCs even when their O:C ratios are close to 1 (Kurten et al., 2016). Please correct.

Lines 16-19: Since Mutzel et al., 2015; several studies have discussed the reactivity of organic hydroperoxides. For example, Surratt and co-workers have reported recently the formation of organosulfates from the acid catalyzed hydrolysis of organic hydroper-oxides (e.g. isoprene dihydroxydihydroperoxides). Please revise this sentence.

Line 24: Please specify the reagent ion used: NO3-

Page 4: Lines 13-14: How does the temperature impact the integrity of the compounds (acids, organosulfates)? Indeed, it has been reported recently that accretion products could decompose at high temperatures (Lopez-Hilfiker et al., 2016).

Page 7: Lines 11-14: The authors should better discuss this point and the potential artifacts of their measurements. The ionization efficiency could be impacted by the

structure of the compounds but also by the composition of the mobile phase (organic phase enhances the ionization efficiency). Is it the reason why the authors decided to use the mobile phases (i.e. water spiked with ACN and ACN spiked with water) described page 6? In addition, the authors should point out the potential matrix effect (i.e. ion suppression). Have they investigated this aspect?

Lines 18-21: Please add references: Glasius and co-workers.

Lines 22-24: Could the authors provide the concentration of isoprene since they also identified some isoprene-derived organosulfates (e.g. m/z 213; 215)?

Page 8: Lines 1-2: I agree that measurements of sesquiterpenes required special setup due to their high reactivity but it is not clear why the oxidation products cannot be measured due to their "high reactivity and low volatility". If they are low volatile the techniques used in this study should allow the detection of such products in gas and particle phases. In addition, could the authors provide the reference(s) reporting the reactivity of sesquiterpene oxidation products? Have the authors compared the products identify in this study with previous works, such as Chan et al. (2011)?

Lines 5-6: The mass accuracy (formula determination) obtained from the UHPLC/ESI-HRMS cannot be solely used to validate the presence of organosulfates. MS2 data of organosulfur compounds such as organosulfate lead to specific fragment ions such as m/z 97 or 80. As recently reported by Riva et al. (2015), other organosulfur compounds such as sulfonate could be distinguished by analyzing the MS2 spectra. Therefore, the authors should use the MS2 data generated from the UHPLC/ESI-HRMS to further support their assignments.

Lines 7-9: This sentence is a bit confusing. Are the contributions of CHON, CHOS and CHONS based on the number of compounds identified? The relative abundance? Please clarify.

Line 28: The authors wrote "since inorganic species are typically not volatilized and

ionized by the AeroFAPA ion source." Please support this statement by a reference.

Lines 33-34: Could the authors further discuss the event identified during the night of the 21st of July? Why are the particles with relative large diameters attributed to a regional source? Did the authors observe any increase on the concentration of inorganic species during that night (sulfate)? Any anthropogenic tracers exhibit a larger concentration?

Page 9 Lines 4-6: This sentence is not clear. What do the authors mean by "deviations between the signals of the instruments"? The authors should also consider the ozonolysis of monoterpenes in the formation of oxygenated species. As recently reported by Yan et al. (2016) ozonolysis of a-pinene is an important pathway in the oxidation of monoterpenes during the night. Finally organic hydroperoxides could also be formed from the oxidation of monoterpenes. Please revise this sentence.

Lines 16-18: How do the authors know that the 93 compounds have a major contribution to the organic aerosol? Indeed, no quantification has been performed and Aero-FAPA-MS is not sensitive (as mentioned by the authors paragraph 2.2) to a large variety of non-acidic compounds such alcohols, hydroperoxides, or aldehydes.

Lines 19-21: Please revise this sentence. It is not because the two data sets correlate that means the Aero-FAPA-MS explains the variability in OA measured by the AMS.

Page 10. Line 30: Further information/discussion should be added in the new version of the manuscript to better justify the choice of the selected organosulfates. Indeed, the choice appears arbitrary and is not well justified. Why did the authors consider only the organosulfates from C7 to C10? Finally the authors should cite the previous laboratory studies that have identified the precursors for the different organosulfates/biogenic tracers identified in this work (c.f. Table S5).

Page 11. Lines 29-34: The authors should estimate the aerosol acidity and correlate the different class of organosulfates vs the aerosol acidity. As written in the manuscript

their results suggest that organosulfate formation is higher under high RH and high concentration of sulfate periods. As reported in previous works the aerosol acidity significantly enhances the formation of organosulfate. However, smaller effects were observed under high-RH and is attributed to dilution of aerosol acidity by additional particle water (Gaston et al., 2014). Therefore, as presented in the manuscript the results appear different than those previously reported and the impact of aerosol acidity should be further discussed.

Page 12. Lines 5-7 and 21-24: Formation of HOMs has been proposed to occur through auto-oxidation reactions, which could lead to the formation of organic hydroperoxides. Recent works have reported the hydrolysis of organic hydroperoxides in the aerosols and the formation of oligomers as well as organosulfates (Lim and Turpin, 2015; Riva et al., 2016a; b). In addition to the proposed reactive pathway, the authors should further discuss the potential formation of organosulfates from the hydrolysis of organic hydroperoxides, especially since they suggest that "aqueous-phase chemistry plays a major role for HOOS production" (Page 11, line 32-34).

Line 15: Why did the authors focus the discussion on only one RO2 radical (i.e. C10H15O8) while they have identified 4 RO2 radicals? Do they have the same profiles? Is the lifetime of RO2 long enough to be transferred to the particle phase and react with sulfate? Is the concentration of C10H15O8 large enough to explain the formation of the parent ion at m/z 327? If the authors expect/propose the RO2 radicals to be the precursors of the HOOS a box model is needed to evaluate the possibility of such reactive pathway.

Finally, the authors should also provide similar time series for the ion at m/z 326 (C10H16O8 + NO3) and compare the time-series of HOMs with the different RO2 radicals.

Please update/correct according to your answers the abstract as well as the conclusion.

References:

Chan et al., 2011; Atmos. Chem. Phys., 11, 1735–1751. Darer et al., 2011; Environ. Sci. Technol., 45, 1895–1902. Gaston et al., 2014; Environ. Sci. Technol., 48, 11178–11186. Hu et al., 2011; Atmos. Chem. Phys., 11, 8307–8320. Kurten et al., 2016; J. Phys. Chem A, 120, 2569-2582. Lim and Turpin, 2015; Atmos. Chem. Phys. 15, 12867-12877. Lopez-Hifliker et al., 2016; Environ. Sci. Technol., 50, 2200-2209. Passananti et al., 2016; Angew. Chem. Int. Ed., 55, 10336-10339. Riva et al., 2015; Environ. Sci. Technol., 49, 5407-5416. Riva et al., 2016a; Environ. Sci. Technol., DOI: 10.1021/acs.est.6b02511. Riva et al., 2016b; Atmos. Chem. Phys., 16, 11001-11018. Schindelka et al., 2013; Faraday Discuss., 165, 237–259. Shalamzari et al. 2014; Environ. Sci. Technol., 48, 12671–12678. Shalamzari et al. 2016; Atmos. Chem. Phys., 16, 7135–7148. Yan et al., 2016; Atmos. Chem. Phys. Disc., DOI:10.5194/acp-2016-367.

---

## Author Comment (AC1) · 30 Nov 2016

Black = reviewer comment

Blue = author response

**Reviewer 1:**

"Brüggemann et al. describe a detailed mass spectrometric analysis of organic aerosols during a field study in a forest in southeast Germany in July 2014. The array and novelty of the gas- and particle-phase analyses completed were impressive. The opportunity to compare trends in HOMs, sulfate, and HOOS is exciting and has the potential to improve our knowledge of organosulfate chemistry. However, much of the discussion is general with more focus on the methods and comparison of those methods rather than the science. I suggest major revisions with focus the results pertaining to advancing our knowledge of biogenic SOA."

> The authors thank the reviewer for taking the time to comment on our manuscript. The authors have taken the reviewers comments into account in a revised version of the manuscript. The details given below, show how each comment was addressed in the revised manuscript.

1) "In particular, the detailed comparison of specific HOMs and HOOS in section 3.2 is exciting and should be expanded and examined in greater detail with additional ion markers examined and compared to the LC-HRMS results."

> As explained in section 3.2, the signals for the representative HOOS were selected based on several strict criteria in order to avoid wrong assignments. Therefore, only the signals that matched the criteria the best were selected eventually for further data analysis. Since the AeroFAPA-MS exhibits only unit mass resolution, it is difficult to unambiguously identify and monitor HOOS from the data of this instrument alone. A comparison to the LC-HRMS data is, thus, necessary. However, most of the signals from the LC-HRMS measurements show additional significant signals on the same nominal mass as the HOOS. Assuming that AeroFAPA-MS is measuring the same compounds, this means that a differentiation between HOOS and other compounds is not possible for such signals. Therefore, the authors believe that examining more ion markers from the AeroFAPA-MS data would lead to unnecessary high uncertainties for the discussed results.

2) "Additional discussion of the methods and comparison of the methods are suggested to be moved to the supplemental."

> In agreement with comments made by referee 2, the authors believe that a discussion and comparison of the methods is necessary, since the AeroFAPA-MS is a rather new and not well-established technique. Therefore, the authors would prefer to keep this part in the main text. However, details of the analysis of filter samples by LC-MS were moved to the Supplemental Material, since these procedures are well-known in the literature.

3) "The abstract can be improved significantly. Much of the abstract reads as a list of methods and types of compounds measured with only vague, general aspects of the results stated, such that the unique findings and implications are not as clear as possible. It would be improved by focusing more on the main details of the scientific results and implications of the study, rather than the analytical aspect of the study. Similar advice should be considered for the conclusions section."

> The authors agree with the referee and the abstract as well as the conclusion section were revised according to the helpful suggestions made.

(Please see the revised manuscript for the changed abstract and conclusions sections.)

4) "The statement on lines 21-24 (page 7) oversimplifies the pathway from emission to SOA formation and does not account for oxidation pathways, reaction kinetics, and reaction yields."

Since MBTCA and 3-carboxyheptanedioic acid are well-known and widely used marker compounds in laboratory as well as field studies for the oxidation of α-/β-pinene and d-limonene, respectively (Jaoui et al., 2006; Szmigielski et al., 2007; Müller et al., 2012), the authors do not see an oversimplification here.

5) "Then on line 28 it is stated that "rather low abundances were found for early-generation oxidation products". What is considered "rather low" compared to expected based on other studies? Do the volatilities of the expected compounds suggest that they should be significantly in the particle phase?"

In order to set the expression "rather low" in context, the authors added the results and the reference of another field studie at the site (Plewka et al., Atmos. Env., 2006). With regards to their volatilities, these compounds can be regarded as semivolatile, meaning a significant fraction should partition into the particle phase. In agreement, compounds such as pinic acid are well-known SOA marker compounds resulting from α-/β-pinene oxidation.

The following sentence was added to the text (P7L32):

**Despite these relatively high mixing ratios for monoterpenes and isoprene, Plewka et al. (2006) already observed that early-generation oxidation products of isoprene and terpenes account only for a small part of the total organic carbon content of the particles at the site. In agreement to this work, similar concentrations in the lower ng·m$^{-3}$ range were found during the F-BEACh study for early-generation monoterpene oxidation products, e.g. pinic acid ($c$ = 4.7 (± 2.5) ng·m$^{-3}$).**

6) "On line 31 (page 7), it is stated that ". . .fast photochemical aging processes, eventually result[ed] in high abundances. . ."; without knowledge of the oxidant concentrations or reference to solar radiation measurements, is difficult to assess this general statement."

The authors are aware of the uncertainties, however, believe that sufficient evidence is given for this statement because of the high abundances of signals for photo-oxidation products (see also the references given in the manuscript). In order to account for these uncertainties, the expression "might indicate" was used in the text (see P7 L31). Moreover, a more detailed discussion of these compounds is also given a bit later in the text when the real-time measurements of the aerosol particles are discussed. Nonetheless, to give more evidence for photochemical processing the authors added the mean solar radiation observed at the site throughout the campaign.

The following sentence was added to the text (P8L6):

**This hypothesis is also in agreement with high solar radiation values observed at the site, typically showing a maximum around midday at an average of 393 (± 15) W·m$^{-2}$. In addition, real-time measurements of the organic aerosol fraction and trajectory calculations also suggest a photochemical source for these compounds, as it will be discussed later on in the text.**

7) "The discussion of organosulfates and related compounds (lines 3-11, page 8) is general, focusing most on the method. However, many papers have been published using UPLC-HR-MS to study organosulfates, so this level of detail about the method is not necessary to include in the main text. I would suggest moving elsewhere (supplemental?) and combining this and the following paragraph."

The authors agree with the reviewer and the description of standard procedures for the LC-HRMS measurements were moved to the supplemental material. Only a small section containing important and specific information was kept in the main text.

8) "Additionally, the list of previous papers on line 14 (page 8) is not comprehensive as currently implied."

Here, the authors' intention is not to give a comprehensive review on studies on organosulfates but just to highlight some studies which are related to the work. Nonetheless, in order to give a broader overview several references were added.

The sentence was changed as follows (P8L24):

**Several of the identified sulfur-containing compounds were already studied in the past and found in field and laboratory studies (Liggio and Li, 2006; Surratt et al., 2007; Surratt et al., 2008; Altieri et al., 2009; Kristensen et al., 2011; Nguyen et al., 2012; Lin et al., 2012; Kristensen et al., 2016).**

9) "Overall, the authors should focus on the scientific results rather than explanation of the methods, information about which should be in the methods and supplemental."

As already mentioned above and in agreement with comments made by Referee 2, the authors believe that a discussion and comparison of the methods is necessary due to the novelty and unique combination of the applied methods.

10) "Are the authors suggesting that the RO2 directly reacts with sulfate in an implied heterogeneous reaction (Page 12, lines 30)? Why does the RO2 need to be the immediate precursor? This would contradict work of Surratt and most others in the consideration of the formation of organosulfates. However, as summarized by Xu et al 2015 (PNAS) in Fig 3, RO2 can be a precursor of organosulfates through the subsequent formation of an epoxide, followed by nucleophilic addition."

As stated in the manuscript the authors think that RO2 might be a direct or indirect precursor (see P12 L27). Due to ambient conditions and the limited dataset it's not possible to discriminate an exact reaction mechanism from the presented measurements. In order to clarify this, the passage was revised as follows (P13L32):

**While the observed coinciding concentration profiles are not unambiguous for the limited available dataset, there might be a certain connection between $RO_2^{\bullet}$ and the observed HOOS. In contrast, time series for a possible closed-shell HOM precursors, such as $C_{10}H_{16}O_{10}$ ($m/z$ 358, $[M+NO_3]^-$), show only a weak agreement with signals for HOOS (Fig. S-14). As previously suggested, e.g. by Kurtén and co-workers (2015), $RO_2^{\bullet}$ contain acylperoxy-functionalities which might possibly undergo a nucleophilic attack by $HSO_4^-$, forming the corresponding HOOS, which has been discussed for closed-shell HOMs earlier by Mutzel et al. (2015). Such a mechanism would explain HOOS formation coupling to $RO_2^{\bullet}$ in the particle-phase and/or at the interface. However, knowledge on the existence of such formation pathways still needs to be much better explored.**

The publication suggested by the reviewer (Xu et al., 2015, PNAS) is very interesting; however, it's mainly focusing on the reactions of isoprene and organosulfates formed in reactions of this compound. However, the focus of this work is on the formation of organosulfates from monoterpenes (as discussed in the text). Therefore, the authors think that the aforementioned publication is only of limited relevance for the manuscript.

Specific Comments:

11-12) "Page 2, lines 2-3: It is unclear what "a good agreement" means here. Page 2, lines 4-5: What aspect of HOOS is implied to be increased here?"

> The abstract was thoroughly revised according to the helpful suggestions made by the reviewers.

13) "Page 2, line 11: Is biomass burning SOA really considered anthropogenic? Wildfire smoke is not necessarily human-caused."

> The authors agree with the referee that biomass burning cannot be solely considered as anthropogenic. Therefore, this expression was removed from the manuscript.
>
> The sentence now reads as follows (P2L5):
>
> **Depending on the source of these VOCs the resulting SOA can be classified as anthropogenic SOA (ASOA), e.g. from fossil fuel combustion, or biogenic SOA (BSOA), e.g. from terrestrial or marine ecosystems (Hallquist et al., 2009; Nozière et al., 2015).**

14) "Page 3, line 8: The phrasing "Yet, until now a comprehensive structural elucidation of HOMs was not possible" implies that the structures, rather than just elemental formulas, of HOMs were determined in this study; this seems misleading."

> In order to clarify that no structural elucidation will be discussed in this work, the sentence was changed as follows (P3L6):
>
> **Although a comprehensive structural elucidation of HOMs was not possible until now, it is assumed that these compounds largely contribute to both particle formation and growth (Riipinen et al., 2011; Donahue et al., 2012; Zhao et al., 2013, Tröstl et al., 2016).**

15) "Page 5, lines 2-3: The statement that PM1 mass closure was achieved is not necessary and is misleading, as it suggests that refractory material did not at all contribute to the aerosol at this site, which is difficult to believe."

> In order to avoid misunderstandings regarding refractory material, the sentence was changed as follows (P5L7):
>
> **The quality control of the data acquired by the AMS was made according to Poulain et al. (2014).**

16) "Section 2.1: What is the forest canopy height at the field site, and what height does the canopy start at? This is needed to understand whether various measurements were located within, below, or above the forest canopy. Also, please provide measurement heights in Sections 2.4 and 2.5. It would be helpful if an overview of measurement locations could be provided in Section 2.1."

> As suggested by the reviewer, the missing information were added to the manuscript. The following phrases were added to section 2.1 to give a better overview of the field site (P4L3):
>
> **The canopy height and displacement height are ~23 m and ~15 m, respectively. A mixture of larch, beech, maple, and pine accounts for the rest of the tree population (Staudt and Foken, 2007). All instruments were arranged closely with inlet heights of 4–6 m above ground and at a distance of less than 10 m. Solely, VOC cartridges were sampled in and above canopy level at a distance of ~200 m from the other instruments.**

17) "Section 2.4: Please provide details regarding the sampling inlet. What are the transmission losses of the HOMs and sulfuric acid through the inlet? What were the uncertainties and limits of detection for sulfuric acid during this study?"

According to the reviewer's suggestions, the passage was revised and more information on the instrument were added.

For details, please see the revised passage (P5L9) of the manuscript.

18) "Section 2.6: What VOCs were able to be investigated in this configuration?"

In this configuration only monoterpenes were quantified. According to the reviewer's comment, the following information were added to the manuscript (P6L10):

**Monoterpenes (i.e. $\alpha$-/$\beta$-pinene, $d$-limonene, $\Delta$-3-carene, camphene) were quantified using authentic standards.**

19) "Page 9, line 17: What fraction of the total OA mass did the 93 compounds comprise?"

In this part of the manuscript, correlations between the data of the AeroFAPA-MS and the AMS are discussed. In a first step, all signals of the AeroFAPA-MS (in the $m/z$ range 150–500) are correlated to the organic compounds that were measured by the AMS – but no quantification of the compounds is intended. Then, in a second step, only the signals of the previously discussed 93 compounds are taken into account for this correlation. Therefore, the authors believe that it is not necessary to discuss the fraction of the total OA here.

Furthermore, the authors did not discuss the fraction of the total OA mass for these 93 compounds in more detail because the exact chemical structures are not available from the measurements performed here. Thus, a quantification by standards is impossible and even a concentration estimation, as it is done for a few marker compounds, using a surrogate compound seems not reliable.

20) "Page 9, lines 29-30: Since both MBTCA and pinic acid were quantified by LC-HR-MS, why can't these concentrations be used to calculate the "aging" ratio, rather than the raw AeroFAPA-MS signals?"

The authors believe that a big advantage of the AeroFAPA-MS is the high time resolution. Using data from the filter sample extracts will lead solely to two data points for each day of the campaign period, while using the AeroFAPA-MS signals provides insights into dynamic changes on the timescale of hours or even minutes. Moreover, as discussed earlier in the manuscript, the AeroFAPA-MS shows a good agreement with the LC-MS data. Therefore, it is believed that an additional aging ratio from filter sample measurements would not add significant value to the manuscript.

21) "Page 9, line 33: By "low altitudes", do you mean within the boundary layer?"

The HYSPLIT trajectories (see Supplemental Material) show that at least some of the trajectories reaching the site were travelling within the boundary layer. In order to clarify this point the sentence was changed as follows (P10L12):

**In agreement, HYSPLIT trajectory calculations reveal that arriving air masses typically traveled several days over land with distances of >1,500 km at low altitudes, often within the boundary layer (Figures S-1 to S-6).**

22) "Page 11, lines 30-32: The authors should consider the work of Xu et al 2015 (PNAS) who examined in detail correlations between organosulfates, RH, and sulfate concentrations. This may improve interpretation of these results."

As already mentioned above, the authors are very grateful for the suggestion of this interesting publication. However, due to the focus on reactions of isoprene in this work the authors believe that it is questionable to apply the same conclusions to the observations discussed in the presented manuscript.

23) "Figure 4: The results of this figure should be discussed in greater detail in the text, taking the LC-HRMS data into greater consideration."

The authors agree with the reviewer and the passage was revised. According to the reviwer's suggestion the LC-MS data were taken into greater consideration and compared to the signals of the AeroFAPA-MS.

Please see the manuscript for the revised passage (P10L22).

24) "Figure 5: This figure could be moved to the supplemental, as the main scientific point is not clear."

The authors agree with the reviewer and the figure was moved to the Supplemental Material.

Technical Comments:

25) Page 3, lines 7-8: Fix reference location.

The reference was fixed.

26) Page 4, Section 2.2: The first two sentences should be combined with the following paragraph to be one full paragraph.

As suggested, the first two sections were combined.

27) Section 2.6: The two paragraphs in this section can be combined to avoid the presence of a two sentence paragraph.

As suggested, the first two sections were combined.

28) Page 6, line 11: Why is the ratio listed as 2:8 instead of 1:4?

As suggested, the ratio was changed to 1:4 (note that this section is now in the Supplemental Material).

29) Page 6, lines 16-17: The phrasing of this sentence does not make sense.

The section was revised and the sentence rephrased (note that this section is now in the Supplemental Material).

30) Page 6, line 24: Given the mass resolving power and tolerance (given on lines 21 and 31), for what m/z range can compounds be unambiguously determined?

The authors think that it is not possible to give an exact answer to this question since it depends on the definition of "unambiguous". As written in the manuscript several assumptions had to be made regarding possible formula assignments. Furthermore, all assignments were within a mass accuracy of 5 ppm. The authors believe that these restrictions allow a quite unambiguous formula assignment for the observed signals, however, in order to avoid misunderstandings, the expression "unambiguous" was removed from the manuscript.

31) Page 6, line 29: The "signal threshold. . .for the detection of significant signals" is stated and in arbitrary units. It would be more meaningful if a S/N threshold were listed.

> The authors agree with the reviewer and the sentence was changed as follows (Supplemental Material, S-1.1):
>
> **The threshold for signal abundance was set to 2.5·10⁶ a.u. (i.e signal to noise ratio ≥3) for the detection of significant signals in the obtained chromatograms after background subtraction by the software.**

32) Page 7, lines 4-6: Similar to the previous comment, there is no context given for "integrated peak areas of >10ˆ7 a.u.". Some relationship to S/N thresholds would be useful here.

> The authors agree with the reviewer and the sentence was changed as follows (P7L5):
>
> **[…] only those signals were selected for the subsequent analysis which showed an integrated peak area of >10⁷ a.u. for at least two separate filter samples, resembling a signal to noise ratio of ≥12.**

33) Page 7, lines 6-7: Does this mean that the previously mentioned 695 compounds (on line 3) were not within a mass accuracy range of +/- 5 ppm? This would be concerning.

> The authors are very grateful for this comment, since the actual value that was set here was +/-2 ppm for the formula assignment. Furthermore, the authors would like to stress that the threshold for the signal abundance (i.e. S/N ratio) was more important for the selection of characteristic compounds, since still a large number of assignments was in the range of +/-2 ppm. Therefore, the sentence was changed as follows (P7L4):
>
> **In order to identify characteristic compounds for the organic aerosol fraction from this relatively large number only those signals were selected for the subsequent analysis which showed an integrated peak area of >10⁷ a.u. for at least two separate filter samples, resembling a signal to noise ratio of ≥12. As an additional criterion, the formula assignment for these signals had to show a mass accuracy in the range of ±2 ppm.**

34) "In addition, it would be useful to provide the m/z range of the identified compounds."

> The following sentence was added to the manuscript (P7L7):
>
> **Eventually, these thresholds led to 93 compounds in the nominal mass range of *m/z* 133–387 which were identified from the data analysis (Fig. 1 and Supplemental Material).**

35) Page 7, line 11: I am guessing that the authors mean "and" instead of "or" here.

> The authors agree and the sentence was changed according to the suggestion.

36) Page 7, lines 11-15: This calibration information could be moved to the methods or supplemental so that the authors can focus on science here. Also, it is not clear here whether all 93 compounds were calibrated with pinic acid or just pinic and terpenylic acid.

> The authors believe that the information on the calibration is very important here since not all compounds were quantified by authentic standards. To clarify that pinic acid served as surrogate for all SOA markers, the sentence was changed as follows (P7L12):
>
> **The concentrations of all SOA marker compounds in PM₂.₅ were estimated using pinic acid as calibration standard.**

37) Table 1: Why are m/z 255 and 357 listed here if they were below the LOQ? Was there any day/night difference observed?

> The authors agree with the reviewer and the signals at m/z 255 and 357 were removed from the table.

38) Page 7, lines 19-21: This suggests that MBTCA is a minor (in terms of yield) pinene oxidation product. Is this correct? Please clarify the comparison.

> Here, it is only said that MBTCA is a later-generation oxidation product. Previous studies showed that it is produced with relatively high yields from oxidation of α-/β-pinene. For example, Müller et al. (2012, ACP) show in chamber experiments that the formation MBTCA can explain up to 10% of newly formed SOA mass.
>
> In order to clarify the expression "later-generation oxidation product" was changed to "major oxidation product". The sentence now reads as follows (P7L21):
>
> **While MBTCA depicts a major oxidation product of $\alpha$-/$\beta$-pinene (Szmigielski et al., 2007; Müller et al., 2012), 3-carboxyheptanedioic acid is a major oxidation product of $d$-limonene (Jaoui et al., 2006).**

39) Page 7, lines 23-24: Were these the only VOCs quantified? This information is not provided in the methods, so it is difficult to fully interpret this result.

> As already stated above (see comment 18), in this configuration only monoterpenes were quantified. However, published data on other VOC emissions from previous field campaigns are available. The following sentence was added to the manuscript (P7L29):
>
> **Beside monoterpene emissions, previous studies at the site have shown that average mixing ratios for isoprene are typically in the range of 0.27–0.50 ppb$_V$. An overview on typically VOC mixing ratios at the site can be found in Klemm et al. (2006) and others (Grabmer et al., 2006; Graus et al., 2006).**

40) Page 8, lines 1-2: Move this statement to the methods section.

> In agreement with the suggestions made by reviewer 3, the passage was changed as follows (P8L9):
>
> **Besides several monoterpene oxidation products, also a marker compound for sesquiterpene oxidation, i.e. $\beta$-nocaryophyllinic acid, could be identified (van Eijck et al., 2013). However, the contribution of sesquiterpene oxidation products on particle composition cannot be estimated here because the observed concentrations were typically below the quantification limits.**

41) Supplemental, Page 1: For clarity, please add a header with the title and author names to the supplement.

> According to the manuscript preparation guidelines of ACP no such header or title is necessary. As written on the ACP website: "Supplements will receive a title page added during the publication process including title ("Supplement of"), authors, and the correspondence email. Therefore, please avoid providing this information in the supplement."

42) Fig S-7, bottom panel caption: For clarity, please indicate the instrument used.

43) Fig S-8, caption: Additional information for interpretation of the figure would be helpful.

> According to the reviewer's suggestion, the caption was revised and additional information were added to facilitate the interpretation of the figure.

44) Fig S-9: A legend would be helpful.

The authors do not see the necessity to add an additional legend since the axes already explain all signals.

---

## Author Comment (AC2) · 30 Nov 2016

Black = reviewer comment

Blue = author response

**Reviewer 2:**

"The manuscript presents interesting new work investigating the composition of organic aerosols using a suite of state-of-the-art mass spectrometric techniques. The results are interesting, but some interpretations are not fully supported by the data in the present version of the manuscript. The development of new instrumentation to investigate organic aerosol composition is important and exciting, but it is necessary to have a deep respect for the possible new artefacts and biases associated with the techniques."

> The authors thank the reviewer for taking the time to comment on our manuscript. The authors have taken the reviewers comments into account in a revised version of the manuscript. The details given below show how each comment was addressed in the revised manuscript.

Specific comments:

1) "Check that the word "identification" is used correctly (according to Nozière et al., 2015) throughout the manuscript."

> According to the reviewer's suggestion the manuscript was checked for the use of the word "identification". The only sentence in which it was used was in the abstract, which was thoroughly revised. The word "identification" is no longer used now.

2-9) "Abstract: The abstract needs a thorough revision to correctly reflect the findings. Page 1 Line 20: How do you "identify a characteristic contributor"? L23-24: I would not characterize concentrations around 10ng/m3 as "high". Please be specific instead. L24. The present data does not support the statement that terpene oxidation products dominate the organic aerosol fraction. The AMS data shows OA mass in the range 2-8 microgram/m3, while only few terpene oxidation products have been identified and quantified with a total concentration of about 1% of this mass. L25: How high? L28: Did the air masses pass areas of high BVOC emissions such as coniferous forests? L32: How do you define "unambigious identification"? Generally authentic standards would be needed to support this. P2 L4: The word "reveal" should generally be avoided in the scientific literature. Here you could use "indicate" instead."

> According to the reviewer's suggestion, the abstract was thoroughly revised. Please, see the new version of the manuscript for all changes made.

10) "P3 L4: Why is important to state that the LC-MS analysis is non-target, since most such analyses would be this? I suggest removing the term throughout the manuscript."

> The term "non-target" is commonly used in the analytical chemistry community to describe a certain data analysis approach of high resolution data. The common approach for LC-MS analyses is to look for specific/known signals, i.e. "targeted", in the obtained chromatograms and mass spectra. In contrast, for a "non-target" analysis no such information is used, but the data is analyzed solely by finding signals which show significant abundances above the background. Due to the huge amount of data that are obtained using high resolution LC–MS, this is only possible using specialized software (see also the Experimental section of the manuscript).

Since the word is in any case not misleading but may even help in understanding the different data analysis approach used here, the authors would prefer to keep this term in the manuscript.

11) "P4L11: Were there any size-selection? A previous paper (Brüggemann et al., EST 2015) describes an SMPS system in front of the AeroFAPA-MS."

In this work no SMPS was used in combination with the AeroFAPA–MS. The authors decided not to use such a size selective system prior to the instrument, since the generation of a monodisperse aerosol might have led to concentrations below the detection limits of the instrument.

12) "P4L3: Are all organic components evaporated at 200C? Will organosulfates evaporate at 200C? What happens to components remaining in the particle phase including inorganic species? The answers to these questions must be given in the manuscript."

To the best of the authors knowledge, inorganic compounds, i.e. salts, are not vaporized in the ionization region of the AeroFAPA–MS. In contrast, laboratory experiments showed that heating the inlet of the AeroFAPA to 200 °C leads to a complete vaporization of secondary organic aerosol particles, which were produced by α-pinene ozonolysis. Nonetheless, due to ambient conditions the authors cannot exclude the possibility of uncomplete vaporization of the aerosol particles. Therefore, the authors decided to use the word "supported" instead of "ensured". The sentence now reads as follows (P4L15):

**Evaporation of organic aerosol components prior to ionization was supported by heating the inlet to 200 °C.**

13) "P5L1: "data quality insurance" - do you mean quality control of the data?"

Indeed, the authors want to make this statement. The sentence was changed as follows (P5L7):

**The quality control of the data acquired by the AMS was made according to Poulain et al. (2014).**

14) "P6L7: The extraction solvent seems relatively polar. Was the extraction efficiency of larger carboxylic acids (such as pinic acid) and organosulfates investigated? Could differences in extraction efficiency have affected the relative proportion between e.g. C7 and C10 organosulfates? "

The authors agree with the reviewer, since differences in extraction efficiencies are an inherent problem of filter analysis by LC–MS. However, since authentic standards for organic aerosol compounds are typically not commercially available and have to be self-synthesized if possible, it is difficult to assess these uncertainties. In order to clarify these uncertainties, the following sentence was added (Supplemental Material, S-1.1):

**It should also be noted that differences in extraction efficiencies and matrix effects might have had a significant effect on the observed signal abundances, which is an inherent problem for aerosol analysis by LC–MS.**

15) "P6L11-12: Is it correctly understood that the average recovery only reflects the loss during evaporation, not extraction efficiency?"

The recovery rate actually also includes losses due to extraction efficiencies. The method description was revised accordingly and now reads as follows (see Supplemental Material):

> **To compensate for losses during the sample processing, i.e. extraction efficiency and evaporation, an average recovery rate was determined for pinic acid, which served as a surrogate for the quantification of other monoterpene oxidation products.**

16) "P7L1- and Table 1: It is impressive that 695 individual compounds are eluted in only 4.1 minutes (according to information on the UHPLC gradient). How were possible matrix effects (leading to signal suppression during ESI) avoided or corrected? "

> The authors agree with the reviewer that it is not possible to exclude possible matrix effects for the LC-MS measurements. However, it should be noted that this is an inherent problem of measurements using LC-MS. The only way to account for matrix effects is to know the composition of the matrix. Unfortunately, this is difficult to achieve (especially for aerosol samples), since the filter samples were taken in the field. In order to clarify this point, the following sentence was added to the detailed method description, which can be found in the revised Supplemental Material:

> **It should also be noted that differences in extraction efficiencies and matrix effects might have had a significant effect on the observed signal abundances, which is an inherent problem for aerosol analysis by LC–MS.**

17) "Why was pinonic acid not quantified? This would have been useful for the discussion of photochemical aging. "

> The authors agree that in several studies pinonic acid concentrations are used to discuss photochemical aging of SOA particles. Therefore, the authors decided to add pinonic acid concentrations to Table 1. However, as suggested by Müller et al. (ACP, 2012), MBTCA is formed from pinonic acid with a yield of only 0.012–1.6% and several other compounds with similar chemical structures might serve as additional precursors for this aging marker. Furthermore, pinonic acid concentrations in the particle phase are easily affected by temperature variations due to the high volatility (SIMPOL.1 model calculated vapor pressure is $5.5 \cdot 10^{-3}$ Pa at 298 K). Therefore, the authors decided to follow the approach of Vogel et al. (EST, 2016) and to discuss the ratio of MBTCA to pinic acid, which is also an early-generation oxidation product of α-/β-pinene but less volatile by a factor of 5 (SIMPOL.1 model calculated vapor pressure is $10^{-4}$ Pa at 298 K).

18) "The higher concentration of MBTCA compared to e.g. pinic acid could indicate that the BSOA components are long range transported rather than locally produced. "

> The authors agree with the reviewer and like to point out that this is already discussed in the original version of the manuscript (see P10 L11): "[…] these extremely high values [of the ratio MBTCA/pinic acid] suggest that mainly air masses with aged aerosol reached the site. In agreement, HYSPLIT trajectory calculations reveal that arriving air masses typically traveled several days over land with distances of >1,500 km at low altitudes, often within the boundary layer (Figures S-1 to S-6)."

19) "It would be useful to compare the concentrations in Table 1 to previous measurements in similar areas. Please remove compounds below limit of detection from Table 1 and just mention them in the text."

> The authors added the results and the reference of another field study at the site (Plewka et al., Atmos. Env., 2006). In agreement with the reviewer's suggestion, compounds below the limit of detection were removed from Table 1.

The following sentence was added to the text (P7L32):

**Despite these relatively high mixing ratios for monoterpenes and isoprene, Plewka et al. (2006) already observed that early-generation oxidation products of isoprene and terpenes account only for a small part of the total organic carbon content of the particles at the site. In agreement to this work, similar concentrations in the lower ng·m⁻³ range were found during the F-BEACh study for early-generation monoterpene oxidation products, e.g. pinonic acid ($c$ = 2.9 (± 2.8) ng·m⁻³) and pinic acid ($c$ = 4.7 (± 2.5) ng·m⁻³).**

20) "P8L11: It is a bit surprising that Hallquist et al. already in 2009 made such general conclusions on the ubiquity of organosulfates and nitrooxy organosulfates, given the very few studies conducted at that time. Please recheck or update the reference."

It is true that at that time knowledge on such compounds was not as detailed as today. However, several sections in Hallquist et al. (2009) are already discussing the abundance and formation of organosulfates and nitroxy organosulfates in ambient aerosols (e.g. p5183–5186). Therefore, the authors believe that it is correct to cite this reference here.

21) "P9L7: Have any of the commercially available standards for organosulfates been analyzed with AeroFAPA-MS? Since the ionization technique is known to form adducts (Brüggemann, Karu and Hoffmann, J. Mass Spectrometry 2016), how was it investigated that organosulfates or organonitrates are not formed during analysis of complex samples such as aerosols containing both organic and inorganic components?"

The authors agree with the reviewer that the offline application of this ionization technique was shown to produce adducts (as discussed in the given reference). However, this adduct formation was only observed to a very small extent for the AeroFAPA (i.e. online) setup (see also Brüggemann et al., ES&T, 2015). Moreover, adduct formation was never observed with sulfate but only with nitrate ions. Therefore, the authors believe that the signals for organosulfates are not affected by such processes in the ionization region. This is also supported by the time trends of AeroFAPA-MS and LC-MS signals for HOOS (see Figure S-9), which show a similar pattern over the campaign period. The signals of the AeroFAPA-MS for organonitrates are not further discussed in the manuscript, since this is not the focus of this work.

22) "P9L12: What do you mean by "quantify"? "

The authors agree with the reviewer that the word "quantify" might lead to misunderstandings here. Therefore, the word "estimate" was used instead. The sentence was changed as follows (P9L25):

**In order to estimate the portion of organic compounds in aerosol particles that was measureable by AeroFAPA–MS, the signals of AMS organics and the TIC of the AeroFAPA–MS were plotted against each other.**

23) "P9L13-26: Just because the two data sets correlate, it does not mean that the Aero-FAPA-MS signal explains the variability in the AMS organic matter data. There could be other underlying common factors involved, such as long-range transport or photochemical processes. "

The authors disagree with the reviewer in this point. It is a widely-used and well-established interpretation of the square of the correlation coefficient (i.e. $R^2$) to take it as a measure for the explained variability of a two-dimensional dataset. As an example, a quote from a course on statistics from Yale University:

"The square of the correlation coefficient, r², […] represents the fraction of the variation in one variable that may be explained by the other variable. Thus, if a correlation of 0.8 is observed between two variables […], then a linear regression model attempting to explain either variable in terms of the other variable will account for 64% of the variability in the data." (see: http://www.stat.yale.edu/Courses/1997-98/101/correl.htm)

More information can easily be found in standard textbooks on statistics (e.g. page 254 in "Statistics Explained", Perry R. Hinton, 3rd Ed., Routledge, 2014).

Of course, the correlation coefficient alone cannot be used to proof causality; however, in this case both instruments were measuring at the same time the same aerosol mass. Moreover, the time series of the instruments also suggest a good agreement. Therefore, the authors believe that causality is given here and the observed correlation gives evidence for a linear association between the observed signals of the two instruments. Furthermore, the authors do not state that this value will give any information on underlying common factors such as the chemical composition of the aerosol mass, which might be influenced e.g. by long-range transport or photochemical processing.

24) "Without any quantification of the Aero-FAPA-MS measurements, the statements that "these compounds reflect major sources" and "particle phase was dominated by BSOA markers" remains not fully documented by the data. Please correct the sentence to reflect your findings more accurately."

The authors agree with the reviewer and the sentences were changed as follows (P9L29 and P10L3):

**Furthermore, the AeroFAPA–MS signals ([M–H]⁻) of the 93 compounds, which were previously identified from LC–MS data as characteristic contributors to the organic aerosol fraction, were plotted as a function of the organic aerosol mass, determined by the AMS (Fig. 3, panel b).**

**As can be seen from this figure, the AeroFAPA–MS spectra support the aforementioned hypothesis that the composition of the particle phase reaching the site were influenced by BSOA marker compounds such as […].**

25) "P9L27: Concentrations of pinonic acid were not listed."

As suggested by the reviewer, the concentration of pinonic acid was added to Table 1 (please see also the answer to comment 17).

26) "P9L28-30: Since MBTCA is an oxidation product of pinonic acid (with OH) the ratio MBTCA to pinonic acid would make more sense. Please include relevant references for using this ratio."

Since not only pinonic acid but also pinic acid is a major early-generation oxidation product the authors think that it is reasonable to use the ratio MBTCA/pinic acid as a proxy for aging processes. This ratio was already used in the literature before (e.g. Vogel et al., ES&T, 2016). Furthermore, due to the higher volatility the partitioning of pinonic acid into the particle phase is stronger affected by temperature variations as it is the case for pinic acid.

The above-mentioned reference was added to the sentence (P10L8):

**The ratio of signals for MBTCA and pinic acid, which can be used as aging proxy for organic aerosols (Vogel et al., 2016), […].**

27) "P10L1: "transported aerosol masses" -> "transported air masses" The text on this page is quite "lengthy" and could be shortened and clarified. "

As suggested by the reviewer, the expression was changed to "**transported air masses**". Moreover, the text of this paragraph was revised and, as suggested by reviewer 1, some additional information were added to facilitate the interpretation of Fig. 4. (Please see the revised manuscript for the new paragraph, P10L5.)

28) "P10L19-24: The extraction efficiency of larger compounds with the polar solvent could also affect the ratio. Since authentic standards of highly oxidized nitrooxy carboxylic acids have probably never been measured by Aero-FAPA-MS, the statement about "reliable detection" seems speculative. Furthermore, possible in-source formation should be investigated. On-line methods are certainly important for measurements of these compounds, but further work is also needed."

In agreement with the reviewer's suggestion, the authors added the following sentence to the method description to clarify possible influences of extraction (Suppl. Material):

**It should also be noted that differences in extraction efficiencies and matrix effects might have had a significant effect on the observed signal abundances, which is an inherent problem for aerosol analysis by LC–MS.**

Furthermore, to weaken the statement of "reliable detection" and to account for yet unknown in-source mechanisms this part was changed as follows (P11L12):

**Thus, online detection methods such as AeroFAPA–MS might allow a more reliable detection of such highly oxidized nitrooxy carboxylic acids in organic aerosols. Nonetheless, different ionization efficiencies and in-source formations might also have a significant effect on the detection of such compounds and should be investigated further in the future.**

29) "P11L3 and Figure 5. The purpose of Figure 5 is not clear. It seems that one compound was not "found" but rather "selected". The figure should be moved to supplementary information. The first paragraph of page 11 could be shortened and focused."

This first paragraph might not directly give the reader additional information on the aerosol composition, but the authors believe that this part is essential for understanding on how signals for HOOS were chosen. As explained in the manuscript, these compounds were indeed "selected" from the LC–MS data set, according to the given criteria. Since referee 3 suggested to give an even more detailed discussion here, the paragraph was not shortened but a few additional information were added. Nonetheless, in order to avoid distraction from the focus of the work, Figure 5 was moved to the Supplemental Material.

30) "P11L7: In agreement with what? "

To clarify this point, the sentence was changed as follows (P11L33):

**This finding is also in agreement with previous studies in which the C7 and C9 HOOS have been identified in laboratory and field measurements by Surratt et al. (2008).**

31) "P11L17-28: Are the HOOS presented and discussed here only the compounds selected in the paragraph above? Please make this more clear - also in the text of Fig. 6. Could the difference between the HOOS classes observed with LC-MS and Aero-FAPA-MS also be due to differences in ionization efficiency and the question of extraction efficiency discussed above?"

In order to clarify that only the four previously identified/selected compounds are discussed, the following sentence was added to the text (P12L10):

**In the following only the signals of these four representative HOOS are discussed.**

32) "P11L32-35: Higher concentration of sulfate (and pH) also affects surface uptake and reactions."

The authors agree with the reviewer and additional information on aerosol acidity were added and compared to time series of different organosulfates. As a proxy for the aerosol acidity molar concentrations of $H^+_{Aer}$ were used, calculated from AMS data as previously shown by Zhang et al. (EST, 2007). In general, no correlation between aerosol acidity and organosulfate formation was observed here. However, it should be noted that the aerosol acidity was quite low over the entire campaign period. In order to give the reader more details on particle acidity and possible effects on organosulfate formation, the following passage was added (P12L28):

**Furthermore, it should be noted that for the entire campaign period the particle acidity was very low and rather stable (average of $H^+_{Aer}$ = 7.4 nmol m$^{-3}$), indicating the presence of partially or even fully neutralized particles (Fig. S-11). In contrast to previous studies, which suggest aerosol acidity to be one of the main factors driving organosulfate formation (Surrat 2007, 2008; Iinuma et al., 2009, Gaston et al., 2014), no such effect was observed here. This result is, however, not contradicting previous findings, but rather indicating that even at low particle acidities HOOS formation can be observed, as it will be discussed in the following.**

33) "P12L8-13 and L17: It is not clear how the hypothesis of "rapid phase transition" of HOMs is supported by the present data. RH is closely related to temperature, and thus time of day, which also affects emissions of BVOC. There could thus be other explanations than condensation for the variation in level of HOMs. It is very interesting how the levels of HOOS and peroxy radicals vary together."

The authors agree that RH is closely related to temperature, and thus, time of day. However, during the discussed period of July 21 the concentrations of HOMs differ strongly from the diurnal cycle which is observed for the rest of the campaign period (see Fig. S-12). RH values reached much higher values during this time (>90% for midday) compared to other days. As shown by Shiraiwa et al. (PNAS, 2011), RH might be a major factor for uptake of gas-phase species into the particle phase. Therefore, the authors believe that phase transition of HOMs was playing an important role for the low HOM concentrations observed. In order to clarify these points, the passage was revised and now reads as follows (P13L9):

**Assuming that RH is a one of the main factors driving the uptake of gas-phase species into the particle phase (Shiraiwa et al., 2011), the required rapid phase transition of gas-phase HOMs is further supported by the observed trend for the sum of HOMs, measured by the CI–APiTOF–MS. In contrast to the diurnal behavior observed for other days of the campaign, the HOM concentrations show very low concentrations during this high humidity period (Fig. S-12).**

34) "P12L28-31: Please clarify how this relates to previously proposed mechanisms for OS formation."

In order to relate the proposed connection between RO2 radicals and organosulfates to mechanisms previously reported, the manuscript was revised and several information on organosulfate formation were added where necessary.

Please see the following comments and corresponding answers for more details: comment 10 by reviewer 1; comment 32 by reviewer 2; comments 2, 19, and 20 by reviewer 3.

35) "Conclusion: Please adjust according to your answers to questions stated in this review."

According to the reviewer's helpful suggestions the conclusion section was thoroughly revised. Please see the manuscript for the new version of this section.

36) "Figure 2 lower panel: The figure is too overloaded with information in overlying graphs. Please make the figure more clear by e.g. moving the data sets further apart. "

The authors agree with the reviewer and the figure was revised.

37) "Figure 5 should be moved to supplementary information."

The authors agree with the reviewer and the figure was moved to the Supplemental Material.

38) "Figure 6. The figure is too busy. Part A could be moved to supplementary information. The marker size is too large for high concentrations of sulfate, which gives a bias in the understanding of the number of data points. Write e.g. C7 vs. C8 to make the figures easier to understand. Are the HOOS all compounds or just the ones selected to represent each group?"

The authors agree with the reviewer and the figure was revised. The marker size was decreased to increase the visibility of data point for higher sulfate concentrations. Moreover, the figure labels in panel a) were changed to C7 vs. C8, etc. To clarify that here only the previously selected HOOS are discussed the caption of the figure was changed as follows:

**Figure 5. (a) Correlations among the selected HOOS signals as well as the effect of RH (color code) and particulate sulfate on their abundance (marker size, range: 0.8–7.2 µg·m$^{-3}$). […].**

---

## Author Comment (AC3) · 30 Nov 2016

Black = reviewer comment

Blue = author response

**Reviewer 3:**

"The manuscript presents interesting investigation on the chemical characterization of organic aerosols collected in southeast Germany. Large variety of mass spectrometry techniques (online and offline) was deployed and the results proposed in this study could provide new insights in the organosulfate chemistry. However, the discussion/interpretation of the results are not well supported and most of the discussion is focus on the comparison of the methods. I suggest major revisions of the manuscript to better discuss the different findings of this study. In particular, the discussion on the potential formation of organosulfates from the heterogeneous oxidation of RO2 radicals is not well sustained and requires deeper investigation. For instance, only one RO2 radical was considered in the discussion (while 4 were identified) and the authors did not consider the other pathways recently reported in the literature (i.e. heterogeneous reactivity of organic hydroperoxides). In general, the authors should better compare/discuss their results with the existing literature."

> The authors thank the reviewer for taking the time to comment on our manuscript. The authors have taken the reviewers comments into account in a revised version of the manuscript. The details given below show how each comment was addressed in the revised manuscript.

1) "Page 2: lines 27-32: The authors mentioned the different pathways leading to the organosulfate formation but some reaction pathways are missing. Indeed, organosulfates could be formed by either nucleophilic substitution of an organic nitrate group by sulfate (Darer et al., 2011; Hu et al., 2011), or by heterogeneous chemistry of gas-phase organic hydroperoxides, which might undergo acid-catalyzed perhydrolysis followed by reaction with sulfate ions (Riva et al., 2016a; 2016b). "

> The authors agree with the reviewer and the following sentence was added to the manuscript (P2L26):
>
> **Furthermore, it was shown that such compounds are formed by nucleophilic substitution of nitrate groups by sulfate (Darer et al., 2011; Hu et al., 2011), or by heterogeneous chemistry of gas-phase organic hydroperoxides, which might undergo acid-catalyzed perhydrolysis followed by reaction with sulfate ions (Riva et al., 2016a; 2016b).**

2) "In addition, the authors should also include the following references: - Reactive uptake of epoxides: Shalamzari et al., 2014; 2016 - Sulfate radicals: Schindelka et al., 2013. -SO2: Passananti et al., 2016"

> The authors thank the reviewer for these additional references. The following sentence was changed as follows (P2L22):
>
> **Studies have shown that OS and NOS are formed in the condensed phase, either from VOC gas-phase oxidation products with sulfuric acid in acidic sulfate aerosols (Iinuma et al., 2005; Liggio and Li, 2006; Iinuma et al., 2007; Surratt et al., 2007; Surratt et al., 2008; Shalamazari et al., 2014; Shalamazari et al., 2016), or also directly by the reaction of gaseous $SO_2$ with unsaturated carboxylic acids (Shang et al., 2016, Passananti et al., 2016).**

3) "Page 3: lines 1-5. It has been recently reported that HOM monomers formed from the oxidation of α-pinene are unlikely to be ELVOCs even when their O:C ratios are close to 1 (Kurten et al., 2016). Please correct."

To account for these recent indings, the following sentence was added to the text (P3L2):

**Nonetheless, recently it was shown that HOM monomers formed from the oxidation of α-pinene are unlikely to exhibit saturation vapor pressures in the range of ELVOCs – even when their O:C ratios are close to 1 (Kurten et al., 2016).**

4) "Lines 16-19: Since Mutzel et al., 2015; several studies have discussed the reactivity of organic hydroperoxides. For example, Surratt and co-workers have reported recently the formation of organosulfates from the acid catalyzed hydrolysis of organic hydroperoxides (e.g. isoprene dihydroxydihydroperoxides). Please revise this sentence."

The authors agree with the reviewer and the following sentence was added (P3L18):

**Nonetheless, recently Riva et al. (2016a) reported on the formation of organosulfates from the acid catalyzed hydrolysis of organic hydroperoxides.**

5) "Line 24: Please specify the reagent ion used: NO3-"

The sentence was changed according to the reviewer's suggestion (P3L23):

**[…] and Chemical Ionization Atmospheric-Pressure interface Time-of-Flight Mass Spectrometry (CI-APi-TOFMS) using nitrate ($NO_3^-$) as ionization reagent (Jokinen et al., 2012).**

6) "Page 4: Lines 13-14: How does the temperature impact the integrity of the compounds (acids, organosulfates)? Indeed, it has been reported recently that accretion products could decompose at high temperatures (Lopez-Hilfiker et al., 2016)."

Unfortunately, at this point the authors cannot completely rule out the decomposition of accretion products during the analysis. However, the authors believe that such contributions to the number and abundance of signals are rather small since a good agreement to AMS and LC–MS measurements is observed. Furthermore, it should be noted that the temperature given in the text (i.e. 200 °C) is solely the temperature of the inlet but not the temperature that is reached for the aerosol flow. Nonetheless, in order to clarify this point the following passage was added to the section (P4L16):

**Although heating is a common approach for aerosol evaporation and analysis, it should be noted here that recently Lopez-Hilfiker et al. (2016) reported on the decomposition of accretion products upon heating. Therefore, contributions of such decomposition products cannot be completely ruled out here.**

7) "Page 7: Lines 11-14: The authors should better discuss this point and the potential artifacts of their measurements. The ionization efficiency could be impacted by the structure of the compounds but also by the composition of the mobile phase (organic phase enhances the ionization efficiency). Is it the reason why the authors decided to use the mobile phases (i.e. water spiked with ACN and ACN spiked with water) described page 6? In addition, the authors should point out the potential matrix effect (i.e. ion suppression). Have they investigated this aspect?"

The authors agree with the reviewer and added information on potential artifacts and ionization efficiencies. For this point, please also read the answer to comment 14 of reviewer 2. Furthermore, the following sentence was added to the main text (P7L15):

**Furthermore, the composition of the LC eluent can have additional effects on the actual ionization efficiencies.**

8) "Lines 18-21: Please add references: Glasius and co-workers."

> The authors do not see any missing references here. Thus, no changes were made. If some essential references are missing here, please specify more clearly.

9) "Lines 22-24: Could the authors provide the concentration of isoprene since they also identified some isoprene-derived organosulfates (e.g. m/z 213; 215)?"

> For the campaign period, no data on isoprene concentrations are available. However, previous studies at the site showed that isoprene mixing ratios are in the range of 0.27–0.50 $ppb_V$ (Klemm et al., Atmos Env, 2006). To give the reader an estimate on the contribution of isoprene to BVOC emissions at the site the following sentence was added to the text (P7L29):
>
> **Beside monoterpene emissions, previous studies at the site have shown that average mixing ratios for isoprene are typically in the range of 0.27–0.50 $ppb_V$. An overview on typically VOC mixing ratios at the site can be found in Klemm et al. (2006) and others (Grabmer et al., 2006; Graus et al., 2006).**

10) "Page 8: Lines 1-2: I agree that measurements of sesquiterpenes required special setup due to their high reactivity but it is not clear why the oxidation products cannot be measured due to their "high reactivity and low volatility". If they are low volatile the techniques used in this study should allow the detection of such products in gas and particle phases. In addition, could the authors provide the reference(s) reporting the reactivity of sesquiterpene oxidation products? Have the authors compared the products identify in this study with previous works, such as Chan et al. (2011)?"

> The authors agree with the reviewer and the statement was changed as follows (P8L10):
>
> **However, the contribution of sesquiterpene oxidation products on particle composition cannot be estimated here because the observed concentrations were typically below the quantification limits.**

11) "Lines 5-6: The mass accuracy (formula determination) obtained from the UHPLC/ESI-HRMS cannot be solely used to validate the presence of organosulfates. MS2 data of organosulfur compounds such as organosulfate lead to specific fragment ions such as m/z 97 or 80. As recently reported by Riva et al. (2015), other organosulfur compounds such as sulfonate could be distinguished by analyzing the MS2 spectra. Therefore, the authors should use the MS2 data generated from the UHPLC/ESI-HRMS to further support their assignments."

> The authors agree that using MS2 data allows to distinguish between organosulfates and compounds such as sulfonates. However, in this case no MS2 data are available. In order to increase the time resolution of the LC-MS method (i.e. to decrease the cycle time of the Orbitrap-MS), which is beneficial in non-target approaches (e.g. Hun et al., 2016, Anal. Bioanal. Chem.), no MS2 data were acquired during the analysis. Nonetheless, to clarify that sulfonates might have a certain contribution here, the following sentence was added to the manuscript (P8L17):
>
> **It should, however, be noted that hydroxysulfonates are isobaric with organosulfates, and thus, might contribute to a certain extent to this class.**

12) "Lines 7-9: This sentence is a bit confusing. Are the contributions of CHON, CHOS and CHONS based on the number of compounds identified? The relative abundance? Please clarify."

> In order to clarify this point the sentence was changed as follows (P8L19):

> **While only 4% of the number of compounds were classified as CHON compounds about 47% of the compounds are either belonging to the CHOS or the CHONS group.**

13) "Line 28: The authors wrote "since inorganic species are typically not volatilized and ionized by the AeroFAPA ion source." Please support this statement by a reference."

> The authors cannot provide a reference for this statement, however, believe that this assumption is rather trivial. Since inorganic species, i.e. salts, can be considered as non-volatile, it seems unlikely that such species can be volatilized in the ionization region which exhibits maximum temperatures of merely 150 °C. Furthermore, a volatilization of inorganic species was never observed by the authors in any measurements.

14) "Lines 33-34: Could the authors further discuss the event identified during the night of the 21st of July? Why are the particles with relative large diameters attributed to a regional source? Did the authors observe any increase on the concentration of inorganic species during that night (sulfate)? Any anthropogenic tracers exhibit a larger concentration?"

> In agreement with HYSPLIT back trajectories (see Supplemental Material) and a strong increase of sulfate during the night (see Fig. 6, panel b) the authors are convinced that mainly particles from regional sources reached the site. To improve the discussion of the data, the following phrase was added (P9L11):

> **This observation is further supported by HYSPLIT backward trajectories, exhibiting rather low altitudes and trajectory lengths (Supplemental Material), as well as a strong increase in sulfate during the night (see also Fig. 5, panel b).**

15) "Page 9 Lines 4-6: This sentence is not clear. What do the authors mean by "deviations between the signals of the instruments"? The authors should also consider the ozonolysis of monoterpenes in the formation of oxygenated species. As recently reported by Yan et al. (2016) ozonolysis of a-pinene is an important pathway in the oxidation of monoterpenes during the night. Finally, organic hydroperoxides could also be formed from the oxidation of monoterpenes. Please revise this sentence."

> To clarify what is meant by "deviations between the signals", the sentence was revised as follows (P9L17):

> **Deviations between the signals for organics of the AMS and the total signals of the AeroFAPA–MS are mostly observed during nightime […].**

> Regarding the reference suggested by the reviewer, the authors do not see how this work fits into the discussed data. While in Yan et al. (2016) gas-phase measurements of HOMs and a subsequent source apportionment are discussed, here the focus lies on the composition of aerosol particles. Moreover, it is well known that ozonolysis of monoterpenes is occurring day and night. Therefore, this reaction cannot explain the deviations observed during nighttime for the AMS and AeroFAPA–MS measurements.

16) "Lines 16-18: How do the authors know that the 93 compounds have a major contribution to the organic aerosol? Indeed, no quantification has been performed and Aero-FAPA-MS is not sensitive (as mentioned by the authors paragraph 2.2) to a large variety of non-acidic compounds such alcohols, hydroperoxides, or aldehydes."

> The authors agree with the reviewer and the expression "major contribution" was changed to "characteristic contribution". Furthermore, these compounds were identified from the

LC–MS measurements and not from AeroFAPA–MS data. In order to clarify, the sentence was restructured as follows (P9L29):

**Furthermore, the AeroFAPA–MS signals ([M–H]⁻) of the 93 compounds, which were previously identified from the LC–MS data as characteristic contributors to the organic aerosol fraction, were plotted as a function of the organic aerosol mass, determined by the AMS (Fig. 3, panel b).**

17) "Lines 19-21: Please revise this sentence. It is not because the two data sets correlate that means the Aero-FAPA-MS explains the variability in OA measured by the AMS."

The authors disagree with the reviewer in this point. It is a widely-used and well-established interpretation of the squared correlation coefficient (i.e. $R^2$) to take it as a measure for the explained variability of a two-dimensional dataset. As an example, a quote from a course on statistics from Yale University:

"The square of the correlation coefficient, r², […] represents the fraction of the variation in one variable that may be explained by the other variable. Thus, if a correlation of 0.8 is observed between two variables […], then a linear regression model attempting to explain either variable in terms of the other variable will account for 64% of the variability in the data." (see: http://www.stat.yale.edu/Courses/1997-98/101/correl.htm)

More information can easily be found in standard textbooks on statistics (e.g. page 254 in "Statistics Explained", Perry R. Hinton, 3rd Ed., Routledge, 2014).

Of course, the correlation coefficient alone cannot be used to proof causality, however, in this case both instruments were measuring at the same time the same aerosol mass. Therefore, the authors believe that causality is given here and the observed correlation gives evidence for the linear association of the observed signals of the two instruments.

18) "Page 10. Line 30: Further information/discussion should be added in the new version of the manuscript to better justify the choice of the selected organosulfates. Indeed, the choice appears arbitrary and is not well justified. Why did the authors consider only the organosulfates from C7 to C10? Finally, the authors should cite the previous laboratory studies that have identified the precursors for the different organosulfates/biogenic tracers identified in this work (c.f. Table S5)."

The discussion of the selection procedure was revised and additional information added to the passage. However, and also in agreement with comment 29 by referee 2, the authors believe that the discussion on the selection criteria of the signals for HOOS should not be extended further in order to avoid distraction from the focus of the work.

Please see P11L21f for the revised passage.

19) "Page 11. Lines 29-34: The authors should estimate the aerosol acidity and correlate the different class of organosulfates vs the aerosol acidity. As written in the manuscript their results suggest that organosulfate formation is higher under high RH and high concentration of sulfate periods. As reported in previous works the aerosol acidity significantly enhances the formation of organosulfate. However, smaller effects were observed under high-RH and is attributed to dilution of aerosol acidity by additional particle water (Gaston et al., 2014). Therefore, as presented in the manuscript the results appear different than those previously reported and the impact of aerosol acidity should be further discussed."

The authors agree with the reviewer and additional information on aerosol acidity were added and compared to time series of different organosulfates. As a proxy for the aerosol

acidity molar concentrations of $H^+_{Aer}$ were used, calculated from AMS data as previously shown by Zhang et al. (EST, 2007). In general, no correlation between aerosol acidity and organosulfate formation was observed here. However, it should be noted that the aerosol acidity was quite low over the entire campaign period. In order to give the reader more details on particle acidity and possible effects on organosulfate formation, the following passage was added (P12L28):

**Furthermore, it should be noted that for the entire campaign period the particle acidity was very low and rather stable (average of $H^+_{Aer}$ = 7.4 nmol m⁻³), indicating the presence of partially or even fully neutralized particles (Fig. S-11). In contrast to previous studies, which suggest aerosol acidity to be one of the main factors driving organosulfate formation (Surrat 2007, 2008; Iinuma et al., 2009, Gaston et al., 2014), no such effect was observed here. This result is, however, not contradicting previous findings, but rather indicating that even at low particle acidities HOOS formation can be observed, as it will be discussed in the following.**

20) "Page 12. Lines 5-7 and 21-24: Formation of HOMs has been proposed to occur through auto-oxidation reactions, which could lead to the formation of organic hydroperoxides. Recent works have reported the hydrolysis of organic hydroperoxides in the aerosols and the formation of oligomers as well as organosulfates (Lim and Turpin, 2015; Riva et al., 2016a; b). In addition to the proposed reactive pathway, the authors should further discuss the potential formation of organosulfates from the hydrolysis of organic hydroperoxides, especially since they suggest that "aqueous-phase chemistry plays a major role for HOOS production" (Page 11, line 32-34)."

The authors agree with the reviewer and are grateful for this suggestion and the corresponding references. The following sentence was added to the passage (P13L7):

**An additional or alternative pathway for the formation of HOOS might be the hydrolysis of hydroperoxide-containing HOMs as recently reported for methylglyoxal-, isoprene-, and alkane-derived hydroperoxides (Lim and Turpin, 2015; Riva et al., 2016a, 2016b).**

21) "Line 15: Why did the authors focus the discussion on only one RO2 radical (i.e. C10H15O8) while they have identified 4 RO2 radicals? Do they have the same profiles? Is the lifetime of RO2 long enough to be transferred to the particle phase and react with sulfate? Is the concentration of C10H15O8 large enough to explain the formation of the parent ion at m/z 327? If the authors expect/propose the RO2 radicals to be the precursors of the HOOS a box model is needed to evaluate the possibility of such reactive pathway."

The authors focused on the discussion of this single RO2 radical (C10H15O8) because it showed the highest abundances during the campaign. In addition, three of the four identified RO2 radicals show very similar time trends and are all possible precursors for the C10 HOOS (i.e. m/z 293 = C10H15O6; m/z 311 = C10H17O7; m/z 325 = C10H15O8). Only the RO2 radical at m/z 393 (i.e. C10H15O10) shows a different behavior. To clarify this point, the time traces of the missing RO2 radicals were added to the Supplemental Material. Furthermore, the passage was revised and now reads as follows (P13L25):

**Remarkably, the signals for three of the four identified $RO_2^\bullet$ (i.e. $C_{10}H_{15}O_6^\bullet$, $C_{10}H_{17}O_7^\bullet$, $C_{10}H_{15}O_8^\bullet$) and HOOS follow the same trends during the dry periods, possibly revealing a connection between these species. Solely the signals for $C_{10}H_{15}O_{10}^\bullet$ (i.e. $m/z$ 357, $[M+NO_3]^-$) exhibit a different behavior (Fig. S-13).**

Although it is argued that there might be a direct or indirect link between these species and HOOS formation, the authors believe and emphasize that further laboratory work under controlled conditions is necessary to discriminate possible underlying reaction mechanisms. The application of a box model with large uncertainties would, thus, only add little additional evidence here and is beyond the scope of this work.

Nonetheless, to clarify that future work is needed to identify underlying formation pathways, the passage was revised as follows (see P13L32):

**While the observed coinciding concentration profiles are not unambiguous for the limited available dataset, there might be a certain connection between $RO_2^{\bullet}$ and the observed HOOS. In contrast, time series for a possible closed-shell HOM precursors, such as $C_{10}H_{16}O_{10}$ ($m/z$ 358, $[M+NO_3]^-$), show only a weak agreement with signals for HOOS (Fig. S-14). As previously suggested, e.g. by Kurtén and co-workers (2015), $RO_2^{\bullet}$ contain acylperoxy-functionalities which might possibly undergo a nucleophilic attack by $HSO_4^-$, forming the corresponding HOOS, which has been discussed for closed-shell HOMs earlier by Mutzel et al. (2015). Such a mechanism would explain HOOS formation coupling to $RO_2^{\bullet}$ in the particle-phase and/or at the interface. However, knowledge on the existence of such formation pathways still needs to be much better explored.**

22) "Finally, the authors should also provide similar time series for the ion at m/z 326 (C10H16O8 + NO3) and compare the time-series of HOMs with the different RO2 radicals."

The authors agree with the reviewer and an additional time series for the HOM species at m/z 326 (i.e. C10H16O8 + NO3) was added to the Supplemental Material. Nonetheless, since the focus of this work is not to investigate connections between closed-shell HOMs and RO2 radicals, the authors compared the time series to the formation of HOOS. Furthermore, Figure S-12 is already comparing the time series for closed-shell HOMs and RO2 radicals over the campaign period. The following sentence was added to the manuscript (P13L34):

**In contrast, time series for a possible closed-shell HOM precursors, such as $C_{10}H_{16}O_{10}$ ($m/z$ 358, $[M+NO_3]^-$), show only a weak agreement with signals for HOOS (Fig. S-14).**

---

## Author Response (AR2)

BLACK = reviewer comment

BLUE = author response

**Anonymous Referee #2**

**Report #2**

The authors thank the reviewer for taking the time to review the revised manuscript. The authors have taken the comments into account and the manuscript was revised where necessary. The details given below show how each comment was addressed.

2.1) In point 12 the authors must at least try to comment on the central question: Do organosulfates evaporate at 200C?

This is also related to the main question in point 21 below.

The authors assume that the reviewer's question is whether organosulfates will completely evaporate at 200 °C during the residence time of particles in the heating and ionization region. As already explained in our first reply, the authors cannot exclude the possibility of uncomplete evaporation of extremely low-volatile particle components, such as organosulfates. Furthermore, it is difficult to give a general answer to this question, since the evaporation behavior will depend on the exact chemical species, its concentration, particle size/phase state, and the matrix (i.e. other compounds present in the particle). However, it should also be noted that in contrast to offline or semi-continuous methods (e.g. FIGAERO-CIMS, MOVI-CIMS), the particles are desorbed directly in air and not from a surface. Thus, a much larger surface area is exposed for thermal desorption. To give an estimate, the following example shows that complete evaporation of organosulfates is typically achieved:

Using the aerosol evaporation model described by Riipinen *et al.* (Atmos. Environ., 2010, 44, 597–607), we calculated the evaporation behavior for 100 nm spherical particles, containing solely organosulfates, in the heating and ionization region for different mass concentrations and temperatures. Values for average saturation vapor pressure ($p_0$) and saturation vapor mass concentration ($C_0$) were taken from Li *et al.* (ACP, 2015, 16, 3327–3344). The parameters used for calculation are given in Table 1 below.

*Table 1: Parameters used for the aerosol evaporation model.*

| variable | value | comment and unit |
|----------|-------|------------------|
| $M$ | 280 | average molecular weight [g/mol] for OS (Li et al., 2015) |
| $\Delta_{vap}H$ | $8.00 \cdot 10^4$ | vaporization enthalpy [J/mol] (estimated by the Joback/Reid method for dodecyl sulfate) |
| $p_{sat}$ | $8.80 \cdot 10^{-7}$ | average saturation vapor pressure at 298 K [Pa] for OS (Li et al., 2015) |
| $\rho$ | $1.36 \cdot 10^6$ | particle density [g/m³] (Riipinen et al., 2010) |
| $D_{i,0}$ | $5.80 \cdot 10^{-6}$ | diffusion coefficient at 298 K [m²/s] (Riipinen et al., 2010) |
| $T_{amb}$ | 298 | ambient temperature [K] |
| $T_{heat}$ | 298 – 498 | heating temperatures [K] |
| $C_0$ | $9.95 \cdot 10^{-8}$ | initial saturation vapor mass concentration [g/m³] (Li et al., 2015) |
| $\sigma$ | 0.06 | surface tension of organic aerosol particles [N/m] (Riipinen et al., 2010) |
| $\lambda$ | $6.20 \cdot 10^{-8}$ | mean free path [m] |
| $t$ | 0.3 | residence time of particles in the heating and ionization region [s] |
| $V$ | $4.9 \cdot 10^{-6}$ | volume of the heating and ionization region [m³] |

Figure 1 shows the mass fraction remaining (MFR) as a function of desorption temperature and ambient mass concentrations. As can be seen from the figure, the calculations suggest that particles completely evaporate during their residence time in the heating and ionization region for all given mass concentrations and temperatures clearly below 200 °C. Although these calculations can only give a first estimates for the evaporation behavior of arganosulfates, they indicate that complete evaporation is typically achieved.

[Figure]

*Figure 1: Calculated mass fraction remaining (MFR) for organosulfate particles as a function of desorption temperature and particle mass concentrations.*

2.2) In point 14 the authors do not adequately answer my question about their extraction efficiency, but instead add a very general statement on this issue. They must at least discuss this point more thoroughly in their reply.

As stated in our first reply, the authors agree with the reviewer on this issue. However, differences in extraction efficiencies for organosulfates and resulting uncertainties cannot be fully assessed here. Nonetheless, it should be noted that typically organosulfates are rather water soluble, especially for higher O:C ratios (e.g. Altieri et al., ACP, 2009; Li et al., ACP, 2016). Since the used extraction solvent was a mixture of water/methanol (1:9), oxidized organosulfates should have been extracted with a high efficiency from the filter samples, reducing errors due to different solubilities. Moreover, it should also be noted that the extraction procedure is very similar to extraction procedures commonly used for organosulfate measurements (e.g. Iinuma et al., EST, 2007; Surratt et al., EST, 2008; Altieri et al., ACP, 2009; Lin et al., ACP, 2012; Nozière et al., Chem. Rev., 2015). Nonetheless, since Riva et al. (ACP, 2016) showed that the selection of the extraction solvent might have a significant impact on the total amount of extracted organosulfates, the following sentence was added to the description of the extraction procedure (Supplemental Material, S-1.1):

**It should also be noted that differences in extraction efficiencies and matrix effects might have had a significant effect on the observed signal abundances, which is an inherent problem for aerosol analysis by LC–MS, as recently discussed by Riva et al. (2016) for measurements of organosulfates from gas-phase oxidation of alkanes.**

2.3) Point 20: Even though Hallquist et al. has written about organosulfates, it is not clear that the review shows that they are ubiquitous, due to the limited number of studies available back in 2009.

The authors agree that Hallquist et al. do not explicitly use the word "ubiquitous" when describing formation and abundance of organosulfates. Therefore, the authors decided to name more recent studies, showing the widespread abundance of organosulfates in ambient aerosols. Accordingly, the sentence was changed as follows (P8L20):

**This large number of organosulfates and nitrooxy organosulfates is, however, not surprising since these compound classes are ubiquitously found in organic aerosol particles and readily accessible for deprotonation via electrospray ionization (Iinuma et al., 2007; Surratt et al., 2007, 2008; Altieri et al., 2009; Hallquist et al., 2009; Schmitt-Kopplin et al., 2010; Gómez-González et al., 2012; Lin et al., 2012; Kahnt et al., 2013; O'Brien et al., 2014; Staudt et al., 2014; Nozière et al., 2015; Riva et al., 2015, 2016a,b).**

2.4) Point 21: The authors " believe that the signals for organosulfates are not affected by such processes in the ionization region". Why was this not investigated using commercially available standards in both the offline and online versions of the instrument? This would normally be a minimum requirement for a new measurement technique.

Although the technique was characterized thoroughly for the analysis of organic aerosols in a large number of experiments, the focus of previous studies was mainly on organic aerosol formation under laboratory conditions without sulfur-chemistry. Moreover, authentic organosulfate standards are not easily accessible. Therefore, the authors cannot exclude adduct formation for organosulfates. However, adduct formation was only observed to a minor extent for other organic compounds (typically <5% of signal for $[M–H]^-$) in the online version of the technique. Furthermore, even if adduct formation occurred to a higher extent, it can be assumed that this would be the case for all organosulfates and, thus, would not greatly affect the conclusions drawn here.

In order to clarify this point, the following sentence was added to the manuscript (P4L22):

**Fragmentation and adduct formation was only observed to a minor extent under laboratory conditions (<5% of signal for $[M–H]^-$), however, such processes might have influenced the observed signals to a greater extent during the field campaign due to ambient conditions.**

As stated previously, the offline version exhibited a higher degree of fragmentation and adduct formation. However, the offline version of the technique was neither used during the field study nor discussed in the manuscript.

2.5) Point 23: I know the statistical background, but the problem (as brought up by reviewer 3 as well) is that the presentation confuses correlation and causation e.g. when the authors write: "supporting the hypothesis that these compounds reflect major sources of the organic aerosol mass at the site." Furthermore, it should be kept in mind that the explanation of variability is for either variable, i.e. AMS-OM also explains the Aero-FAPA-MS signal. When two distinguished reviewers bring this up, it should be a hint to the authors to write the statements more clearly.

The authors agree that the statement could lead to confusion and misunderstandings and should be revised. Although the AMS is, of course, not free from measurement uncertainties, this instrument is taken as a reference here, since it represents a well-established technique for aerosol analysis. The assumption behind the phrase "supporting the hypothesis that these compounds reflect major sources of the organic aerosol mass at the site" is that the AMS is giving the sum of (almost) all organic substances, whereas only 93 signals were taken into

account from the AeroFAPA–MS. As explained in the manuscript, these 93 signals show almost exactly the same behavior as the sum of signals for organics from the AMS. The authors agree that this doesn't mean that these compounds are abundant in a large quantity, nonetheless, they reflect the general behavior of the organic aerosol mass at the site. In addition, these 93 compounds were also detected at elevated levels from the filter analysis by LC–MS.

In order to clarify this section, the revised version now reads as follows (P9L34):

**Similar to the correlation of the TIC of the AeroFAPA–MS to the organic aerosol mass, a linear correlation was found for the 93 signals ($R^2$ = 0.80). Taking the well-established AMS as a reference and neglecting measurement uncertainties, this correlation indicates that about 80% of the organic aerosol's variability can be explained by these 93 signals, supporting the hypothesis that these compounds reflect the general behavior of the organic aerosol fraction at the site.**

2.6) Abstract: In the new version of the abstract the following sentences should be clarified according to the discussion (and to make them easier to read):

[revised manuscript text omitted]